

# Physical-chemical properties of particles in hailstones from Central Argentina

Anthony C. Bernal Ayala[1], Angela K. Rowe[1], Lucia E. Arena[2,3], and William O. Nachlas[4]

[1]Department of Atmospheric and Oceanic Sciences, University of Wisconsin, Madison, WI, USA
[2]Facultad de Matemáticas, Astronomía, Física y Computación, Universidad Nacional de Córdoba, Córdoba, Argentina
[3]Observatorio Hidrometereológico de Córdoba, Córdoba, Argentina
[4]Department of Geoscience, University of Wisconsin, Madison, WI, USA

**Correspondence:** Anthony C. Bernal Ayala (crespo3@wisc.edu)

**Abstract.** This study presents a novel analysis of two hailstones collected in central Argentina to provide insights into the size distribution, composition, and potential sources of non-soluble particles within hailstones. Using this new method, non-soluble particles are trapped beneath a thin layer of polyvinyl resin and analyzed with Confocal Laser and Scanning Electron Microscopy combined with Energy-Dispersive Spectroscopy, preserving their in-situ location and physical characteristics. The study characterized these particles' distribution, shape, and size and identified their elemental composition, which is used to interpret possible source regions. Particles ranged in diameter from 1.2 to 256.0 microns, with the largest found in hailstone embryos. Agglomerated mineral and organic particles dominated the elemental composition, followed by organics and quartz, and were present throughout the hailstones. Agglomerated salt particles detected in one sample were traced to a nearby salt lake, while copper chloride and zinc chloride particles found in the second sample were potentially associated with agrochemicals commonly used for pest control and fertilizer, including in Argentina. Various local and regional land-use types, including shrublands, mixed vegetation, croplands, and urban areas, were linked to specific types of particles. This study, therefore, highlights the regional influence of various land use types on hail formation and growth, pointing to the potential impacts of natural and anthropogenic factors on hailstone composition.

## 1 Introduction

Falling hailstones are some of the most destructive natural phenomena in convectively active regions of the world. In the U.S. alone, hailstorm damage has exceeded 1 billion dollars since 1949 (Changnon, 2008; Sander et al., 2013; Allen et al., 2017; Kumjian et al., 2019) and can significantly impact the local economy, particularly in agricultural regions. It is crucial to study hail-producing storms from a global perspective owing to remaining uncertainties in the environmental conditions associated with hail-producing storms of varying modes. However, this task is often limited by the lack of confirmed reports of hail and their properties. Studies have explored how environmental factors affect hail production in deep convective storms through high-resolution modeling (e.g., Kumjian and Lombardo, 2020) and using global inferences of hail occurrence through satellite-based proxies (Cecil and Blankenship, 2012; Ni et al., 2017; Bang and Cecil, 2019; Bruick et al., 2019); however, they often are not verified against hail observations. Even when operational radar data is available with accompanying hail reports, accurately



estimating hail sizes from these observations remains challenging (e.g., Murillo and Homeyer, 2019). Much remains unknown
about the processes leading to hail growth and environmental controls on hail occurrence and size, particularly outside of the
United States (Allen et al., 2020).

Ice nucleating particles (INP) are a subset of environmental aerosols that facilitate ice formation through heterogeneous
freezing owing to the assistance of an insoluble particle such as mineral dust derived from surface sediments, biological
material (e.g., pollen, bacteria, fungal spores, and plankton), and volcanic ash (Lamb and Verlinde, 2011; Vali et al., 2015).
While a range of aerosol types can serve as INP (Lamb and Verlinde, 2011; Zhao et al., 2019; Burrows et al., 2022), various
factors affect particles' ice-nucleating capabilities, including their size, surface topography, and composition including coating
(e.g., Holden et al., 2021; Gao et al., 2022) and even less is known about the types and concentrations of aerosols involved in
hail formation. The complexity of deep convective storm microphysics, limited observations, and the shortcomings of current
models contribute to the inconclusive understanding of the effects of INP on hail formation and growth.

Studies of collected hailstones worldwide have provided some insight into potential INPs involved in hail formation.
Michaud et al. (2014) found biological ice nuclei in hailstone embryos in the U.S. Rocky Mountains, while hailstones col-
lected in Slovenia (Šantl-Temkiv et al., 2013) and the triple border region of Paraná, Brazil, and Argentina (Beal et al., 2021)
noted signatures of the respective regions' soil. Others point to anthropogenic markers through the presence of microplas-
tics (Kozjek et al., 2023), highlighting the implications of human activity on hail formation. The analysis techniques in those
studies all required melting the hailstones, removing information on particle size distribution or composition with respect to
the hailstone embryo, and neglecting non-soluble particles. This present work, using a novel hailstone particle analysis method
(Bernal Ayala et al., 2024b), adds to the limited literature on hailstones' composition through analysis of individual non-soluble
particles contained within hailstones collected in Central Argentina, a global hotspot for hail.

Córdoba Province in Argentina has some of the most intense storms in the world (Zipser et al., 2006) and experiences
frequent hail (e.g., Cecil and Blankenship, 2012; Rasmussen et al., 2014) that has destructive impacts on property and agricul-
ture. As such, this region has been the focal point of recent field campaigns collecting data on these severe storms, including
the 2018-19 Cloud, Aerosol, and Complex Terrain Interactions (CACTI; Varble et al., 2021) and 2018 Remote Sensing of
Electrification, Lightning, and Mesoscale/Microscale Processes with Adaptive Ground Observations (RELAMPAGO; Nesbitt
et al., 2021) near the Sierras de Córdoba (SDC; Figure 1), a mesoscale mountain range that runs parallel to the Andes in
northwest Argentina. A 7-month (austral spring to mid-autumn) survey of CACTI INP measurements (Testa et al., 2021) found
that diverse plant communities in nearby areas likely release high amounts of biological and organic particles. Additionally,
intensively farmed plains are identified as a significant source of soil dust from post-harvest to late spring, with a combination
of bio-particles released by plants and soil dust when crops start growing thus contributing to INP concentrations during this
time of year. This CACTI study did not focus on hailstorms, although it is therefore reasonable to hypothesize that hailstones
collected in this region would contain particles containing imprints of both biological and agricultural activity.

To explore the influence of particle identity and availability on hailstone formation, microscopy techniques can be applied
to the individual non-soluble particles using the new method described in Bernal Ayala et al. (2024b). Studies over the past
two decades (e.g., Sokolik and Toon, 1996; Tegen and Lacis, 1996; Claquin et al., 1999; Sokolik and Toon, 1999; Kandler





et al., 2007; Nousiainen et al., 2009; Jeong and Achterberg, 2014; Kemppinen et al., 2015; Li and Sokolik, 2018; Conny et al.,
2019; Huang et al., 2020; Di Biagio et al., 2020) have used Scanning Electron Microscopy (SEM) with Energy-Dispersive
X-ray Spectroscopy (EDS) to identify and quantify the mineral phases present in dust aerosols. Much of this work has fo-
cused on dust samples from desert regions like Asia and the Sahara, identifying key mineral phases including quartz, calcite,
halite, and hematite (Jeong and Nousiainen, 2014; Schütz and Sebert, 1987). The mineralogical complexity of dust particles
is further compounded by amorphous silica, particularly in Saharan dust, often associated with clay minerals (Jeong et al.,
2016). These findings provide a valuable reference framework for interpreting EDS spectra and identifying potential mineral
phases in dust samples. However, to fully understand the role of these particles in ice nucleation, it is necessary to investigate
their mineralogical composition, physical characteristics, and other factors, including agglomeration of multiple particle types.
Previous studies suggest that organic-rich soil dust more efficiently serves as INP than mineral desert dust (Conen et al., 2011;
Steinke et al., 2016; Cornwell et al., 2024), but how much of this enhancement is owing to the presence of organic coatings and
biological material remains unclear. With agricultural soil dust and biological particles likely being primary aerosol sources
available for hail formation in Central Argentina (e.g., Testa et al., 2021), it is crucial to explore the mineral composition of
individual particles to understand potential INP sources for hailstone formation.

   This study uniquely studies the physical and chemical properties of non-soluble particles retaining their position within each
hailstone layer for two hailstorms of varying modes in Central Argentina. The overall motivation of this work is to provide
insight into which non-soluble particles were likely involved in the initial stages of ice nucleation, and thus hail formation
and subsequent growth, as well as their potential source regions. The specific objectives of this study are to (1) characterize
the distribution and size of non-soluble particles trapped in both hailstone samples, (2) identify the elemental composition
distribution of individual non-soluble particles collected by each hailstone during its growth in the cloud, and (3) determine
possible source regions of non-soluble particles identified in the hailstones. Section 2 describes the datasets and methods used
to address these objectives. Section 3 describes the physicochemical characterization of non-soluble particles found in the
hailstone sample. Section 4 discusses the results in the context of previous work and the implications of those results.

## 2  Data and Methods

### 2.1  Hailstone Collection and Preparation

The hailstones used for this analysis were collected during two hail events: 8 February 2018 and 13-14 December 2018; the
later occurring during the RELAMPAGO-CACTI observation period. More specifically, the 4-$cm$ hailstone from 8 February
(hereafter referred to as V-7) was collected in Villa Carlos Paz, Córdoba, Argentina (lime-green star in Figure 1), and the 8-$cm$
hailstone from 13-14 December (hereafter referred to as NG-1) was collected in Villa del Dique, Córdoba, Argentina (blue star
in Figure 1). These collections were facilitated through Lucia E. Arena at the Facultad de Matemática, Astronomía, Física y
Computación at the Universidad Nacional de Córdoba (FAMAF-UNC) and the citizen science "Cosecheros" program (Arena
and Crespo, 2019; Cos, 2022). Collected hailstones were then processed at the FAMAF-UNC's subzero facility, Laura Levi
Atmospheric Physics Laboratory, at a temperature of $-12 \pm 2°\ C$. At this stage, a novel approach to preserve non-soluble





particles within hail layers was implemented (Arena, 2024; Bernal Ayala et al., 2024b). Hailstone samples were attached to a glass base and then cut over the equatorial symmetric plane using a diamond-embedded cutting disk. After cutting the hailstone to expose the embryo, the sample was evenly polished with a microtome to provide a thin, even ice layer over the embryo. At

this stage, pictures were taken to record the location of growth rings relative to the embryo. Immediately after, 1% polyvinyl formal (Formvar) solution diluted in ethylene dichloride was applied with a glass rod over the flat polished surface and left to curate for a few minutes. Once covered with the Formvar solution, the hailstone was left in a sealed low-humidity container with silica gel at $-12°\,C$ to sublimate. Once the ice sublimated, the particles trapped in the Formvar could be analyzed using light and electron microscopy at room temperature while preserving their location with respect to the hail embryo. More details

on these hailstone collection and preparation procedures are provided in Bernal Ayala et al. (2024b).

### 2.1.1   Microscopy analysis

With particles and their locations with respect to the embryo preserved in Formvar, this method then allows for the application of microscopy techniques to provide information on the physical and chemical characteristics of non-soluble particles within hailstones. Using an OLYMPUS LEXT OLS4000 Confocal Laser Scanning Microscope (CLSM), a 2-D cross-section of each

of the V-7 and NG-1 hail samples (Figure 2 and Figure 3, respectively) was created by identifying the embryo and using it as a reference point for scanning regions within and around it. In the case of the 4-$cm$ V-7 sample, the 2-D cross-section (Figure 2) covers the embryo (S4, S5) and outer layers in both directions from the embryo to the outer-most layers of the stone (S1-S3, S6-S8). Within this 2-D cross-section, individual sectors of equal size (i.e., labeled S1-S8 in V-7, Figure 2) were selected for higher magnification to identify individual particles quasi-randomly within each sector with respect to the embryo. A similar

approach was implemented for NG-1 (Figure 3); however, owing to the larger size of this stone (8-$cm$ compared to the 4-$cm$ size of V-7), a second CLSM sweep was required to examine the entire area encompassing the embryo. This additional sweep resulted in horizontally adjacent sectors (e.g., S1, S2) in the same vertical section that were then grouped under a similar layer number for analysis (e.g., in Figure 3, S1 and S2 represented the same layer of the stone and thus identified as "1" when analyzing particles in that area). These numbered layers cover the larger embryo (1-5) and a cross-section toward one end of

the stone identified as the outer layers (6-10).

This approach provided particle size distribution and surface topographical information for 73 and 211 identified particles within the V-7 and NG-1 hailstones, respectively, including particles down to the technique-limited minimum size of $1\mu m$ (Bernal Ayala et al., 2024b). This lower limit is achievable due to the CLSM optical sectioning capability, pinhole aperture design, and high-resolution imaging, which allow for clear visualization of structures approaching the diffraction limit of light

without breaking it. It is worth noting that this resolution differs from the SEM, which can achieve resolutions down to $50\,nm$ due to its use of electrons rather than light, offering even finer detail for nanoscale structures. Filmetrics (2017) software was used to calculate particle size, which is determined as the maximum length along the x- and y-axes. The increase in particle numbers analyzed between the samples is attributed to the increased efficiency in the analysis, allowing for more particles to be examined in a similar amount of time.





The sublimated hailstone samples were then coated with gold (V-7) and gold and palladium (NG-1), the difference owing to the availability of pure gold during lab analysis times, and subsequently analyzed using a Zeiss Sigma field emission gun (FEG) SEM with an EDS X-Max $80mm^2$ detector. Secondary electron images were acquired at 15 $kV$ and a working distance of approximately 8.5 $mm$; settings that dictated the clarity and detail of the imagery and allowed for the identification of heavy metals without disintegrating the non-soluble particles (Bernal Ayala et al. 2024). The 2-D elemental cross-sectional

maps (Figures 2, 3) were used to locate the same particles observed in the CLSM to investigate the elemental composition of the same identified individual particles. To minimize interference from the sample substrate, EDS spectra were acquired for 120 seconds from the center of each particle using a single-point analysis technique. Particle-free areas of the glass were also measured using EDS to obtain a control spectrum from the coated glass substrate. Additional details on this process and considerations when choosing coating material and EDS analysis techniques are available in Bernal Ayala et al. (2024b).

### 135   2.1.2   Elemental characterization

A unique feature of this technique is its capacity to link the elemental composition of individual particles with their location and proximity to the hailstone embryo. Due to these particles' small size and complexity, EDS spectra were primarily used for element identification, focusing on the presence and relative abundance of key indicator elements to interpret particle identity. However, quantitative elemental abundances from EDS measurements are unreliable for several reasons, including

the particles' small size, heterogeneous composition and structure, and the presence of analytical artifacts. Each particle was characterized by its elemental weight percentages (e.g., Laskin et al., 2012), with uncertainties stemming from several factors. The particles were covered by a ~1$\mu$m thick layer of Formvar (a carbon-based material) and further coated with a 10-$nm$ layer of either gold (Au) or a gold-palladium (Au-Pd) conductive coating. Additionally, the particles rested on a soda-lime glass substrate containing elements like silicon (Si), aluminum (Al), sodium (Na), calcium (Ca), potassium (K), and oxygen (O),

which contributed to the X-ray signals. As a result, spectrum deconvolution was necessary to account for signatures from both the coating materials and the substrate, and this was achieved using NIST DTSA-II software (Ritchie, 2010; Bernal Ayala et al., 2024b).

Analyzing particles with irregular geometries presents additional challenges (Fletcher, 2011; Goldstein et al., 2017). EDS is fundamentally designed for flat, homogeneous materials when applied to small, polyphase, or irregularly shaped particles.

Complications arise due to assumptions in the matrix correction and irregularities in X-ray production, emission, and take-off angles to the detector. These factors can significantly impact the elemental abundances calculated from the EDS spectrum, reinforcing the rationale for using EDS for elemental identification rather than for quantifying absolute concentrations.

Analytical artifacts inherent to the EDS detector further complicate the analysis. Issues such as sum peaks, peak overlaps, and incorrect peak identification can lead to erroneous results. Automated spectrum identification often yields numerous incorrectly

identified peaks, as observed in our samples where aluminum (Al) was interfered with by bromine (Br) and titanium (Ti) by barium (Ba). If such element identifications are accepted without scrutiny, many falsely identified peaks could be incorporated into the dataset, rendering the data meaningless.



Additionally, the EDS spectrum represents all X-rays produced within an approximate 2-4 $\mu$m$^3$ activation volume at the beam's impact location. Given that many aerosol dust particles exhibit heterogeneity at the sub-micron scale (as noted in Section 1), numerous EDS spectra will reflect polyphase materials. For these reasons, we did not place high confidence in the absolute values of elemental weight percentages. Instead, EDS spectra were interpreted based on the presence of characteristic elements corresponding to specific mineral phases. Spectra showing elemental abundances indicative of polyphase materials were classified as either lithic fragments (i.e., pieces of eroded rock rich in Si, Al, Na, Ca, Mg, and Fe, hereafter referred to as lithics) or as agglomerated carbon-lithic or carbon-lithic-clay fragments (rich in C, Si, Al, Na, Ca, Mg, and Fe) representing the potential organic mixture with dust and other particles in the region, hereafter referred to as agglomerated mineral/organics.

This elemental classification scheme, which accounts for the aforementioned limitations, identified several distinct minerals, including quartz, calcite, halite, and Fe-oxide. Particles with high concentrations of carbon (C) and oxygen (O) were interpreted as organics, while others with a significant presence of nitrogen (N) were identified as nitrates. We also detected particles containing unique indicator elements, such as groups of zinc chloride, copper zinc chloride or copper chloride particles, with EDS spectra rich in Zn-Cl, Zn-Cu-Cl or Cu-Cl, respectively.

### 2.1.3 Air Mass Trajectories

The NOAA Air Resources Laboratory's Hybrid Single-Particle Lagrangian Integrated Trajectory model (HYSPLIT; Stein et al., 2015) was first used to generate a 24-hour air mass back-trajectory using $0.28° \times 0.28°$ European Environment Agency Reanalysis datasets (ERA5; Hersbach et al., 2020) with a vertical resolution of 37 pressure levels: 25 $hPa$ intervals from 1025 to 750 $hPa$, 50 $hPa$ intervals from 750 to 300 $hPa$, and 25 $hPa$ intervals from 275 to 100 $hPa$. Trajectories were initiated at 1700 $UTC$ on 8 February and 2200 $UTC$ on 13-14 December, with hourly intervals starting at heights of 100, 500, 1000, and 1500 $m$ above ground level from convective core coordinates on 8 February (64.75° W, 31.59° S, marked by a red star in Figure 1) and on 13-14 December (64.45° W, 32.20° S; marked by a blue star in Figure 1). These initiation coordinates were identified using channel 11 (8.4 $\mu$m) from the geostationary satellite GOES-16 (Bernal Ayala et al., 2022).

These initiation height levels were chosen for the following reasons: 100 $m$ is the lowest point to the surface, and the first pressure level, 500 $m$ and 1000 $m$ provide boundary layer variability, and 1500 $m$ focuses on possible particle transport influences from the low-level jet (Bernal Ayala et al., 2024b; Sasaki et al., 2024). A trajectory matrix with a 7x5 grid and $0.30°$ spacing was also processed from the initial convective pixel coordinate at 1700 and 2200 $UTC$ (on 8 February and 13-14 December, respectively) for five days to better understand which local and longer-range sources could have transported particles at the initiation location before hail-producing convection was observed. The area covered by all back-trajectories resulting from the matrix was divided into grid cells with dimensions of $0.28°$ longitude and $0.28°$ latitude (i.e., ERA-5 horizontal resolution). Each trajectory occurrence in each grid cell was then normalized based on the time spent over each grid cell and included trajectory endpoints for all the heights (ERA5; Ashbaugh et al., 1985). Residence-time coefficient pixels were overlaid on the C3S Land Cover classification gridded map from 2023, as shown in Figure 1. This map provides a global description of the land surface divided into 22 classes, available through the C3S Climate Data Store and defined using the United Nations Food and Agriculture Organization's Land Cover Classification (Copernicus Climate Change Service, 2019). This approach





provides insight into the highest probability of specific land-use regions being possible source regions for the non-soluble particulates analyzed in this study.

# 3 Results

## 3.1 8 February 2018

On 8 February 2018, an isolated supercellular convective system developed east of the Sierras Grandes, located in the northern section of the SDC (lime green star; Figure 1). This convective system produced record-breaking gargantuan hail (Kumjian et al., 2020) in Villa Carlos Paz, in addition to the 4-$cm$ hailstone collected and analyzed in this study (V-7). While a warm, humid, conditionally unstable airmass was present, supporting the initiation and growth of deep convection on this day (Kumjian et al., 2020), with a large-scale pattern similar to composites for supercell environments in this region (Mulholland et al., 2018), a northerly low-level jet, often linked to large-scale moisture transport (e.g., Sasaki et al., 2024) and deep, widespread convective systems in this region (e.g., Rasmussen and Houze, 2016), was lacking during this case. This particular hail-producing storm, therefore, developed under more local influences, including strong upslope flow along the SDC (Bernal Ayala et al., 2022), that are likely to affect potential source regions of particles found within the V-7 hailstone from this case.

### 3.1.1 Particle Size and Elemental Composition of Individual Particles

This V-7 hailstone analyzed in this case included particles analyzed both in the embryo and outer layers through a vertical cross-section of the 4 $cm$ stone (Figure 2). The CLSM analysis of V-7 revealed particle sizes ranging from 1.9 to 150.3 $\mu$m (Figure 4), with an average particle size of 40 $\mu$m and a precision uncertainty of $\pm0.2$ $\mu$m, with a minimum size of 1.9 $\mu$m close to the minimum observable particle size using this method (1 $\mu$m; Bernal Ayala et al., 2024b). The standard deviation for the particle size distribution in V-7 is 25.9 $\mu$m. When focusing on particles within the embryo, sizes ranged from 6.4 to 150.3 $\mu$m, with a standard deviation of 30.2 $\mu$m. In contrast, particles in the outer layers were overall smaller over a narrower range, encompassing sizes from 1.9 to 121.8 $\mu$m with a standard deviation of 23.2 $\mu$m. The comparison between the embryo and outer layers in V-7 suggests that the embryo contains larger particles than the outer layers, although both regions contained particles exceeding 100 $\mu$m.

With a large range of particle sizes identified in the hailstone, the classification scheme described in Section 2.3, was applied to this hailstone sample to determine the compositions of these individual particles. Overall, the particles in V-7 were primarily characterized as agglomerated minerals/organics ( 41%; Figure 5-A) associated with C-rich clays and lithics. The particle sizes in this category ranged from 1.93 to 75.24 $\mu$m (Table 1) and were evenly distributed throughout the hailstone (Figure 6), with no notable concentration differences between the embryo (Sectors 4-5) and the outer layers (Sectors 1-3, 6-8).

The second most prevalent group was agglomerated salts ( 16%; Figure 5-A), characterized by strong Cl and Na signals in EDS analysis. Distinct peaks for Cl Na indicate the presence of halite or similar salt minerals. The particle sizes in this category ranged from 5.30 to 150.3 $\mu$m (Table 1), with the largest particle recorded in this group located in the embryo of





the hailstone sample. Interestingly, even though this large particle was identified as a potential salt, the backscatter imagery did not display the typical cubic crystalline structure expected of halite. Instead, this particle exhibited a strong carbon peak,
indicating a small salt particle situated atop a potential agglomerated mineral/organic particle. This observation was consistent across most particles categorized in this group. Agglomerated salts were distributed throughout the hailstone (Figure 6), with no notable concentration differences between the embryo (Sector 5) and the outer layers (Sectors 1-3, 5, 6).

At 8% frequency, Zn-Cl-rich particles (Figure 5-A) consisted of particles with a Zn-Cl content greater than 1% by weight. The particle sizes in this group ranged from 6.89 to 33.49 $\mu$m (Table 1), and they were dispersed throughout the hailstone
(Figure 6), with no evident higher concentration in the embryo (Sector 5) compared to the outer layers (Sectors 3, 6-7). Organic particles ( 8%; Figure 5-A), containing more than 60% weight of carbon and rich in oxygen, were also identified with a similar concentration as Zn-Cl-rich particles, over a similar size range (9.69 to 33.41 $\mu$m; Table 1), and similarly spread throughout the hailstone (Figure 6). Similarly, at 8% frequency, clays (Figure 5-A), characterized by high aluminum content and the presence of Si, K, Ca, Na, Mg, and Fe, were also located both in the embryo (Figure 6; Sectors 4-5) and the outer layers (Sectors 2, 7)
with sizes ranged from 3.98 to 54.61 $\mu$m (Table 1).

Shifting to the larger-than-average particle sizes found in V-7 are quartz particles ( 7%; Figure 5-A, identified as silicon content greater than 30% by weight, with sizes ranging from 15.93 to 54.61 $\mu$m (Table 1), and ammonia/nitrate particles ( 7%; Figure 5-A), which were nitrogen-rich with content greater than 7% by weight and ranged in size from 30.96 to 58.43 $\mu$m (Table 1). Both of these particle categories were also distributed across the hailstone. Conversely, Fe-oxides ( 3%; Figure 5-A).
dominated by iron and oxygen with no carbon or silicon and ranging in size from 33.2 to 91.75 $\mu$m (Table 1), appear to have been exclusively located in the outer layers of the hailstone (Figure 6; Sectors 3, 8). Although Fe-oxides were only detected in the outer layers in this particular swath, a second swath conducted in a different direction across the hailstone (not shown) identified Fe-oxide particles in the embryo as well. This trend for all particle categories to be found throughout the whole sample extended to the final identified category, lithics ( 1%; Figure 5-A), characterized by a silicon content of 20-40% weight,
along with high concentrations of Na, Mg, and Ca, minor amounts of Al and K, and no detectable carbon. Only one particle, measuring 4.94 $\mu$m (Table 1), was identified in this category and located in the hailstone sample's outer layer (Figure 6; Sector 1); however, the second swath (not shown) revealed additional lithic particles within the embryo.

By analyzing the elemental composition of individual particles within the hailstone, the dominance of large, agglomerated particles (including those with salt) in the sample suggests a link to local land use, including hypothesized agricultural and
biological sources. While particles of up to 100 $\mu$m are likely to remain suspended in the atmosphere for up to 2 days (Jaenicke 1978, Bakan et al. 1987), the largest of the particles observed within the V-7 hailstone (i.e., 150 $\mu$m) was likely ingested into the storm from local sources near the SDC. The HYSPLIT back trajectory analysis provides further insight into these links.

### 3.1.2 Possible Source Regions

The analysis of the 24-hour HYSPLIT back trajectory (Figure 8-A) provides information on the possible origins of the particles
in the V-7 hailstone sample with distinct patterns at different altitude levels. Levels 100 and 500 $m$ AGL show trajectories near the initiation point from the northwest but, earlier in the 24-h period, had curved over the northeastern part of the SDC, likely



owing to the upslope flow of surface winds from the northeast (Bernal Ayala et al., 2022). This upslope flow is supported by the terrain and height change of the trajectory seen in Figure 8-A (bottom figure). The 1000 and 1500 $m$ AGL trajectories came from the north and northwest near the initiation point but, moving back in time, showed a slight curvature north and northeast

of the Cordoba Province at distances farther than the lower levels. Within this short time frame, the SDC will likely impact the particle sources found in the hailstone. However, longer range transport of particles before 24-h may also have deposited particles in the area that could have been ingested into the 8 February supercell.

To investigate potential source areas further back in time, the residence time coefficients calculated for the 5-day back-trajectory matrices (Figure 8-B) similarly highlight the regional influence indicated in the 24-h trajectories in that most tra-

jectories lie within the Argentinean borders. More specifically, under the assumption that the particles arriving at the location where the hail-producing storm initiated are more likely emitted from regions where the air masses spent more time (Yadav et al., 2021; Ren et al., 2021; Testa et al., 2021), the grid cells showing the high residence-time coefficients (any grid that includes more than one HYSPLIT trajectory) are considered potentially principal sources for the particles found in the hailstone. The regions with the highest residence-time coefficients for 5-days leading up to the V-7 storm initiation (orange, yellow,

and red in Figure 8-B) are generally located over the SDC, Córdoba City (white star in Figures 1, 8-B), Argentina's largest natural salt lake (Laguna Mar Chiquita; $30.71°$ S, $62.56°$ W), and Provinces such as Santiago del Estero, Chaco, Santa Fe, and Corrientes (see Figure 1 as a geographical reference). These results reveal that sources within Argentina's geographical limits account for possible non-soluble particle sources in the analyzed hailstone sample. More specifically, in comparing the high residence-time coefficient pixels from the 5-day back trajectory with the C3S Land Cover map (Figure 8-B), we find the most

predominant land uses (Figure 8-C) corresponding to shrublands, croplands, mixed vegetation, urban areas (mostly Córdoba city), and areas with a body of water (including the aforementioned salt lake), consistent with the imprints of soil dust, organics, salt, and minerals found throughout the V-7 hailstone, including the embryo, and emphasizing the importance of local land use on INP sources of hail formation.

## 3.2 13-14 December 2018

It is not uncommon for hail-producing storms in Argentina to form from larger convective systems in environments characterized by air transport from non-local sources (e.g., Rasmussen and Houze, 2016; Sasaki et al. 2022, 2024). Therefore, it is worthwhile to compare the V-7 hailstone with one collected in this type of environment to explore potential non-local sources on hail formation. The 8-cm NG-1 hailstone collected during the 13-14 December 2018 event was associated with a mesoscale convective system (MCS) that developed in the southern portion of the SDC (blue star; Figure 1). In contrast to the 8 Febru-

ary supercell case, this hail-producing MCS was associated with increased moisture from the South American Low-Level Jet (SALLJ), dry air subsidence east of the Andes capping the low-level moist layer, and lee cyclogenesis resulting from the passage of an upper-level trough (Bernal Ayala et al. 2022; Sasaki et al. 202240,43); all characteristics of environments favorable for MCSs in this region (Rasmussen and Houze, 2016; Mulholland et al., 2018; Bruick et al., 2019; Sasaki et al. 20249,42,44). As such, particularly owing to the SALLJ's large-scale transport of air and likely associated particles from a farther distance,



the hypothesis was that there would be different particle compositions in this stone compared to the 4-cm stone of 8 February (a supercell dominated by more local influences including strong upslope flow).

### 3.2.1 Particle Size and Elemental Composition of Individual Particles

For NG-1, particle size ranged from 1.2 to 256.0 $\mu$m (Fig. 9), with an average particle size of 35 $\mu$m and a standard deviation of 34 $\mu$m. The largest particle in NG-1 was 100 $\mu$m larger than the largest in V-7, and the average particle size was also higher

in NG-1 (35 $\mu$m) compared to V-7 (30 $\mu$m), as well as having a larger standard deviation. This shows that both hailstone's particle sizes had large variations. In the case of NG-1, the large particles (i.e., > 100 $\mu$m) were found throughout the hailstone, with large size variability observed both in the outer layer and in the embryo. Similar to V-7, the elemental classification scheme applied to NG-1 showed that the particles were primarily composed of agglomerated minerals/organics ( 36%; Figure 5-B), including one of the particles exceeding 100 $\mu$m (Table 1). Also similar to V-7 is that these agglomerates were found

throughout the NG-1 hailstone sample (Figure 9), with no preferred concentration between the embryo (Layers 1-5) and the outer layers (Layers 6-10).

While one relatively small lithic particle was identified in V-7 (Table 1, Figure 6), lithics were the second most prevalent group in NG-1 ( 22%; Figure 5-B), with particle sizes up to 180.5 $\mu$m (Table 1). NG-1 showed a consistent distribution of lithics throughout the hailstone, with the largest particle found within the embryo (Figure 9; Layer 4). Organic particles were

also more prevalent in NG-1 ( 15%; Figure 5-B) than in V-7 ( 8%; Figure 5-A). This category contained the largest particle size found in NG-1 (256.0 $\mu$m; Table 1), located in the embryo (Figure 9; Layer 9), although, like V-7, these particles were dispersed throughout the whole hailstone. Similar to V-7, quartz and clays (Figures 5-B, 9) were also found in NG-1 distributed throughout the hailstone, shifting to slightly larger particle sizes (Table 1). While less prevalent than quartz and clays (Figure 5-B), ammonia/nitrate and Fe-oxide particles were also detected in NG-1, with ammonia/nitrate mostly located in the outer

layers (Figure 9; Layers 6, 8-10), consistent with V-7's findings.

Despite the consistency in particle elemental categories found within both hailstones, some categories were unique to NG-1. Cu-Zn-Cl-rich particles ( 4%; Figure 5-B), with particle sizes ranging from 2.6 to 78.1 $\mu$m (Table 1), were primarily located in the outer layers (Figure 9; Layers 6-10). Cu-Cl-rich particles ( 1%; Figure 5-B), with particle sizes ranging from 9.0 to 33.7 $\mu$m (Table 1), were instead mostly found in the embryo (Figure 9; Layers 3, 5). Calcite particles ( 2%; Figure 5-B), with smaller

particle sizes ranging from 8.4 to 24.1 $\mu$m (Table 1), were distributed in both the embryo (Figure 9; Layers 3-4) and the outer layers (Layers 7, 9). Finally, a single brass particle ( 1%; Figure 5-B) measuring 1.2 $\mu$m (Table 1), characterized by strong Cu and Zn signals in EDS analysis, was found in the outer layer (Figure 9; Layer 7).

Overall, while both hailstone samples showed similarities in certain categories such as quartz, clays, nitrate particles, and Fe-oxides, there were notable differences in the presence and distribution of lithics, organics, Cu-Zn-Cl-rich, Cu-Cl-rich particles,

brass, and calcite, as well as the presence of salts in V-7 but their absence in NG-1. Were these different particle compositions linked to different trajectories associated with these different storms, including more non-local sources on 13-14 December? To explore this question, the HYSPLIT trajectories are again analyzed for the 24-hour and 5-day time frames.





### 3.2.2 Possible Source Regions

The 24-hour HYSPLIT back trajectory analysis was applied to the 13-14 December case (Figure 10-A) to determine potential

source regions of the particles found within the NG-1 hailstone sample. Nearest the initiation point, the trajectories originate from the west and northwest at all analyzed levels. In particular, trajectories curved around the northern section of the SDC, moving upslope along the western side of the southern section of SDC instead of the eastern side as on 8 February. The 1500 $m$ AGL levels followed a similar path as the lower-level winds but started at regions farther north than the lower trajectories. This trajectory height and direction is likely linked to the SALLJ and thus could have led to more remote sources of particles

over a longer time period.

Therefore, residence time coefficients for 5-day back-trajectories matrices were again calculated (Figure 10-B) to explore potential sources outside of Argentina's geographical limits. Unlike on 8 February, when the trajectories were primarily within Argentinian territories, on 13-14 December, air mass trajectories with high residence times also came from Paraguay and Brazil. Under the same aforementioned assumption that the particles arriving at the location where the hail-producing storm

initiated are more likely emitted from regions where the air masses spent more time, the regions with the highest residence-time coefficients (orange, yellow, red and lighter shades of blue in Figure 10-B) were generally located northwest of the SDCs, in the boundary between Cordoba, the southwestern corner of La Rioja and San Luis Provinces (Figure 1) as well as in locations outside of Argentina. While the main land-use categories associated with the highest residence times are generally the same for both cases (i.e., Figures 7c, 10c), the distribution shifts on 13-14 December from pixels associated with shrublands to more

pixels associated with both cropland and mixed vegetation (Figure 10-B). The dominance of the cropland category again is consistent with the prevalence of agglomerated mineral/organics, but the trajectories from either the west side of the SDC or from regions north of Argentina could explain unique source regions to NG-1 resulting in particle elemental compositions not observed or more prevalent than in V-7 (e.g., lithics, Cu-Zn-Cl-rich, Cu-Cl-rich, brass, calcite). There were also low residence time pixels over the salt lake (Figure 10-B), potentially explaining why no salt was found in NG-1.

## 4 Discussion

Our study presents the first analysis of the sizes, composition, and distribution of non-soluble particles in hailstones from storms in Argentina. Unlike previous studies that melted hailstones to evaluate their overall composition (Šantl-Temkiv et al., 2013; Michaud et al., 2014; Beal et al., 2021), our approach (Bernal Ayala et al., 2024b) preserves both the non-soluble particles and their in-situ locations within the hailstones. It uniquely allows for analysis of the elemental composition of individual particles,

revealing elemental combinations that impact the ice nucleating ability of those particles.

Applying our method to two hailstone cases from different storm modes in Central Argentina revealed that the majority of particles in both hailstones are agglomerated minerals/organics, found both within the embryo and throughout the outer layers of the hailstones and exceeding 100 $\mu$m in size. The largest particle (256 $\mu$m in NG-1) was identified in the embryo and characterized as organic, composed primarily of carbon (76%) and oxygen (22%). The presence of organic particles

within hailstones aligns with a recent study of hailstones in China (Zhang et al., 2023), which used SEM-EDS and machine



learning to analyze insoluble particles through a melting and filtering technique. However, our unique approach, which enables the analysis of individual particles, suggests that particle agglomeration may have played a significant role in hail formation within our Central Argentina cases. Indeed, with a link to cropland through our particle trajectory analysis and prior studies of agricultural soil dust emphasizing mineral and organic compositions of wind-eroded sediments (e.g., Steinke et al., 2016; Iturri
et al., 2017; Cornwell et al., 2024), this work supports that organic-mineral soil dust agglomerates likely serve as efficient INP for hail formation.

This link of hail formation to regional soil signatures has also been observed in hailstones collected in Slovenia and the triple border region of Paraná, Brazil, and Argentina (Šantl-Temkiv et al., 2013; Beal et al., 2021), expanding beyond agricultural activities to other influences of local land cover. Their results are consistent with our findings that quartz, clays, and agglom-
erated mineral/organic particles were abundant in our hailstones, particularly associated with the highest residence times of trajectories over and near the SDC. These particle trajectories also showed a link to a nearby salt lake (Laguna Mar Chiquita) on 8 February, where the largest particle in V-7 was categorized as an agglomerate that contained salt. A recent modeling study of this lake emphasized that it is one of many that are shrinking, exposing dry lakebeds that present a source of Na-rich mineral dust (Borda et al., 2022). The presence of this large mineral-organic agglomerate with salt in the hailstone embryo and the
potential link back to this shrinking salt lake emphasizes another anthropogenic source of potential INPs or insoluble CCNs to consider in hail formation and growth.

This anthropogenic link is also suggested through elemental composition results unique to the NG-1 hailstone. In particular, trajectory analysis for this case showed sources both within and outside Argentina under the general "croplands" category identified as the dominant land-use type, thus focusing our interpretation of results on cropland activities over a broader region.
Notably, the presence of Cu-Cl in the NG-1 sample suggests a potential link to agrochemicals like copper chloride, commonly used for pest control (Lewis et al., 2016). According to a market report, copper oxychloride is used globally as a fungicide for crop protection, including in Argentina (Manoj, 2023). The rarity of minerals containing Cu and Cl adds complexity to their origin story. While these minerals are relatively rare, searches on international databases (e.g., mindat.org) reveal several Cu-Cl evaporite minerals. Some of these minerals are actively mined in open-pit mines in the Chilean Atacama Desert, including
the type locality of clinoatacamite. This finding suggests a possible regional evaporite source, although the possibility of agrochemical origins remains compelling.

The abundance of zinc in both samples is also worth highlighting. While zinc is typically not found in significant atmospheric concentrations, it can be introduced into the environment through various processes. In the atmosphere, zinc often appears as an aerosol in an oxidized form and can be transported over long distances via atmospheric deposition. In Argentina, the presence
of zinc in the hailstones is particularly relevant due to the country's significant zinc production. The Aguilar Mine in Jujuy Province produced over 23,000 tons of zinc in 2017 (, USGS; Administration, 2023). Zinc is also commonly used in the agricultural sector as a fertilizer (Tox, 2005), including liquid zinc-based fertilizers (e.g., ZINC 700 and Status ZN), which are used for seed treatment in crops like fruit trees, vineyards, vegetables, rice, corn, and wheat (Argentina; Rizobacter). Natural processes also contribute to the environmental presence of zinc; in particular, Ecandrewsite ($ZnTiO_3$) and other ilmenite-group
minerals have been identified in amphibolites from the SDC basement. Zinc is relatively widespread in this region's associated



metasedimentary and meta-igneous sequences, a finding confirmed by Espeche and Lira (2022) through an electron microscopy technique similar to the technique used in this study.

Additionally, iron was found in the hailstone samples, with three potential sources: natural processes, agricultural activities, and anthropogenic factors in the region. The SDC range likely contributes to atmospheric iron through weathering and erosion. Zaccarini et al. (2004) described the presence of metamorphosed iron ore in the Sierra de Comechingones in Córdoba Province. The iron is typically found in the form of magnetite ($Fe_3O_4$, 72.4% Fe), hematite ($Fe_2O_3$, 69.9% Fe), goethite (FeO(OH), 62.9% Fe), limonite ($FeO(OH) \cdot n(H_2O)$, 55% Fe), or siderite ($FeCO_3$, 48.2% Fe). Weathering and erosion of these iron-bearing minerals can release iron particles into the atmosphere. In addition to the natural contribution from the mountains, agricultural areas around Córdoba may also act as a significant source of airborne iron. Soils in these regions can be naturally rich in iron, which becomes airborne through agricultural activities or wind erosion. The extensive agricultural practices in Córdoba, which include the use of fertilizers containing iron as a micronutrient (e.g., Caburé; S.R.L.) further contributes to atmospheric iron and may release iron-containing dust particles into the atmosphere when applied.

Understanding the origins of these minerals highlights the environmental impact of human activities, such as agricultural practices, and the natural contributions from surrounding geological features, like the SDC, on aerosol sources, including those involved in hail formation and growth. It emphasizes the need to explore further the effects of aerosol sources on hailstone growth and development with a global mindset to improve near- and long-term forecasting of these impacts.

## 5 Conclusions

Using a novel hailstone-analysis technique (Bernal Ayala et al., 2024b), this study provides first-of-its-kind insights into the size distribution, composition, and potential sources of non-soluble particles within hailstones from South America. These hailstones were collected in central Argentina from different storm modes under various environmental conditions, offering an opportunity to investigate potential variability in particle source regions and characteristics within hailstones that could influence their formation and growth. Through this unique approach, we found:

- Analyzed particles ranged in size from 1.2 to 256.0 $\mu$m, with the largest particles in each hailstone found in its embryo and larger particle sizes for the larger of the two hailstones.

- Agglomerated mineral/organic particles dominated both hailstones' elemental composition, an organic particle being the largest, and were observed in all layers throughout the hailstones.

- Lithics and agglomerated salt particles were the next dominant elemental composition found both in the outer layers and in the embryo of V-7 and NG-1, respectively. Size ranges overlapped those of agglomerated mineral/organic particles, suggesting their possible role as INPs for hail formation.

- Contributions from agricultural practices and geological features were linked to hailstone formation. Both hailstones were associated with various regional land-use types, including shrublands, mixed vegetation, croplands, and urban





areas. The particles found contained regional soil signatures, potentially influenced by agrochemicals used in pest control practices or crop fertilizers, as well as sodium-rich mineral dust from nearby dry lakebeds.

These findings enhance our understanding of the interplay between atmospheric particle sources, both natural and anthro-
pogenic, and their role in hailstone formation by linking the composition of particles within the hailstone embryo to local and regional land use. The similarities observed between the two hailstones studied suggest that the composition and size of aerosols, key inputs in hail formation and growth models, are crucial for accurate simulations. However, the larger NG-1 hailstone displayed a greater diversity and number of particles, indicating potential differences in particle accumulation and transport processes, likely influenced by varying storm modes.

This study contributes to ongoing hail research and highlights the potential impact of natural and human activities on hail growth. Nonetheless, one limitation of this technique is the inability to distinguish between organic and inorganic carbon in C-rich particles. Some studies interpret C and O-rich particles analyzed with SEM-EDS to be organic particles, especially true when C and O are not associated with significant amounts of other elements that would indicate inorganic compounds (e.g., Niemeyer, 2015; Avula et al., 2017; Hayden et al., 2023). However, using EDS for this purpose is challenging due
to several factors. Carbon contamination within the SEM chambers can result in low-level carbon detection across all EDS spectra. Moreover, EDS is less sensitive to light elements such as C. While our samples were coated with a C-based Formvar, Bernal Ayala et al. (2024b) demonstrated minimal contribution from the coating in EDS spectra. Additionally, the EDS detector captures C X-rays. Still, it cannot differentiate the bonding environment of the carbon, making it impossible to determine if the carbon originates from inorganic sources (e.g., graphite), organic matter (e.g., detritus or pollen), or industrial materials (e.g.,
soot). As a result, using EDS alone to categorize C-rich particles as biological or non-biological is not reliable, motivating the need for complementary spectroscopic and microscopy techniques, as discussed in Bernal Ayala et al. (2024b).

Another limitation to consider is the minimum observable particle size (Bernal Ayala et al., 2024b). Despite this limitation, the large particles found in our hailstones suggest processes such as coagulation, which likely affect their ice-nucleating abil-
ities. These phenomena are likely not unique to this study area. The techniques used in our research, examining individual
non-soluble particles in hailstones, provide a path forward for analyzing individual particles' size range and elemental com-
position in a variety of environmental and land-use regions. Future studies can, therefore, apply this technique to hailstone samples from different regions globally to investigate whether similar particle characteristics are observed elsewhere. Such re-
search would contribute to our understanding of hail formation mechanisms and help refine predictive models. As we continue to improve modeling efforts, incorporating diverse aerosol compositions and sizes will be crucial for enhancing the accuracy
of forecasts related to hail events.

*Data availability.* The physical (CLSM) and chemical (EDS) data of the two analyzed hailstones are stored in an Excel sheet, which is accessible through Zenodo (Bernal Ayala et al., 2024a). We obtained the ERA5 reanalysis data from Hersbach et al. (2023) and the land cover classification gridded map from the Copernicus Climate Change Service, Copernicus Climate Change Service (2019). This ERA5 data was input to the National Oceanographic and Atmospheric Administration's Hybrid Single-Particle Lagrangian Integrated Trajectory



(HYSPLIT) model (version 5.3.0; available at https://www.ready.noaa.gov/HYSPLIT.php) to produce residence time coefficient plots based on the air mass trajectories.

*Author contributions.*  Conceptualization was carried out by A.C.B.A. and L.E.A.; methodology was developed by A.C.B.A., A.K.R., L.E.A., and W.O.N.; validation was performed by A.K.R., L.E.A., and W.O.N.; formal analysis was conducted by A.C.B.A.; the investigation was undertaken by A.C.B.A., A.K.R., and L.E.A.; resources were provided by A.K.R. and L.E.A.; L.E.A. managed data curation; the original

draft was prepared by A.C.B.A.; writing, review and editing were done by A.C.B.A., A.K.R., L.E.A., and W.O.N., visualization was handled by A.C.B.A.; supervision was provided by A.K.R., L.E.A., and W.O.N.; and funding acquisition was secured by A.K.R. All authors have read and agreed to the published version of the manuscript.

*Competing interests.*  The contact author has declared that none of the authors has any competing interests.

*Acknowledgements.*  We acknowledge the National Science Foundation funding support for this project (AGS-1640452 and AGS-1661768).

We would also like to thank the LAMARX-UNC operators for their assistance in operating the microscopes used in this study, especially Sebastian Garcia, for his assistance in initial assistance during the development of the technique used in this study.



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





| Category | Particle size range ($\mu$m) | |
|---|---|---|
| | *V-7* | *NG-1* |
| *Agglomerated Min/Org* | 1.93 – 75.24 | 3.9 – 108.9 |
| *Ammonia/Nitrate* | 30.96 – 58.43 | 23.21 – 35.85 |
| *Brass* | —- | 1.18 |
| *Calcite* | —- | 8.41– 24.06 |
| *Clays* | 3.98 – 54.61 | 2.83 – 68.60 |
| *Cu-Cl rich* | —- | 32.00 |
| *Cu-Zn-Cl rich* | —- | 5.80 – 78.06 |
| *Fe-oxide* | 33.2 – 91.75 | 20.74 – 61.38 |
| *Lithics* | 4.94 | 4.81 – 180.5 |
| *Organics* | 9.68 – 33.41 | 8.74 – 256.00 |
| *Quartz* | 15.93 – 54.61 | 12.29 – 81.31 |
| *Agglomerated Salts* | 5.30 - 150.3 | —- |
| *Zn-Cl rich* | 6.89 – 33.49 | 2.56 – 33.72 |
| *Cu-Zn-Fe rich* | —- | 26.60 |

**Table 1.** Particle size ranges for each elemental category identified in hailstone samples V-7 and NG-1.





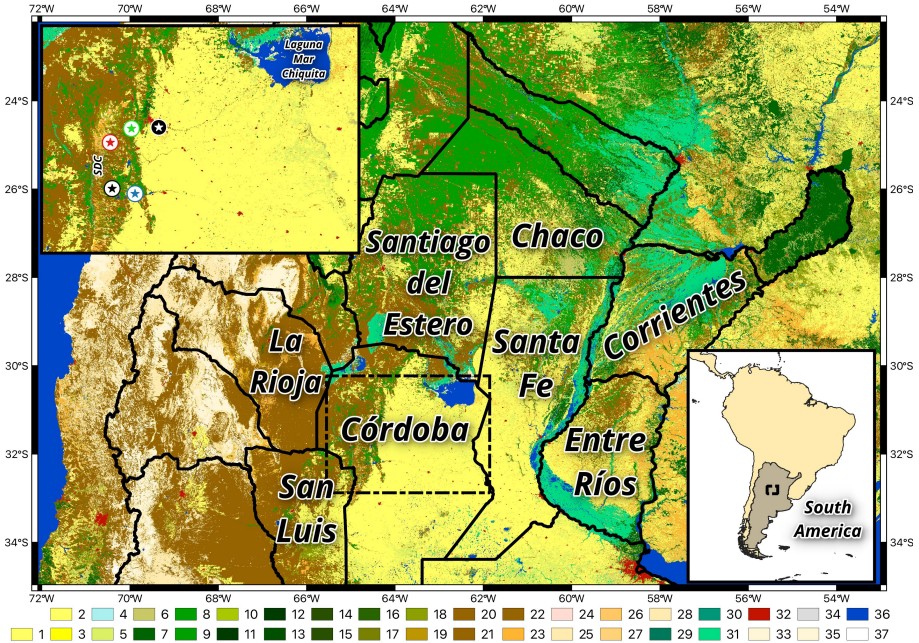

**Figure 1.** Map of northern Argentina covering an area shown in the black box in the lower right panel, including the Córdoba study area and nearby provinces. Within the dashed box covering northern Córdoba province is an inset (top left) highlighting Córdoba City to the east of the Sierras de Córdoba (SDC) and points of interest for this analysis: the site of the CACTI experiment observations in Villa Yacanto (black star), the hail collection locations: Villa Carlos Paz, Villa del Dique (lime green and blue stars, respectively), and the initiation point for air mass back trajectory run for 8 February and 13-14 December (red and blue star, respectively). Color fill represents the C3S Land Cover map available through the C3S Climate Data Store (CDS): 1-cropland rainfed, 2-cropland rainfed, 3-cropland rainfed tree or shrub cover, 4-cropland irrigated, 5-mosaic cropland, 6-mosaic natural vegetation, 6-tree broadleaved ever- green closed to open, 7-tree broadleaved deciduous closed to open, 8-tree broadleaved deciduous closed, 9-tree broadleaved deciduous open, 10-tree needleleaved evergreen closed to open, 11-tree needleleaved evergreen closed, 12-tree needleleaved evergreen open, 13-tree needleleaved deciduous closed to open, 14-tree needle- leaved deciduous closed, 15-tree needleleaved deciduous open, 16-tree mixed, 17-mosaic tree and shrub, 18-mosaic herbaceous, 19-shrubland, 20-shrubland evergreen, 21-shrubland deciduous, 22-grassland, 23-lichens and mosses, 24-sparse vegetation, 25-sparse tree, 26-sparse shrub, 27-sparse herbaceous, 28-tree cover flooded fresh or brackish water, 29-tree cover flooded saline water, 30-shrub or herbaceous cover flooded, 31-urban, 32-bare areas, 33-bare areas consolidated, 34-bare areas un- consolidated, 35-water, 36-snow and ice.





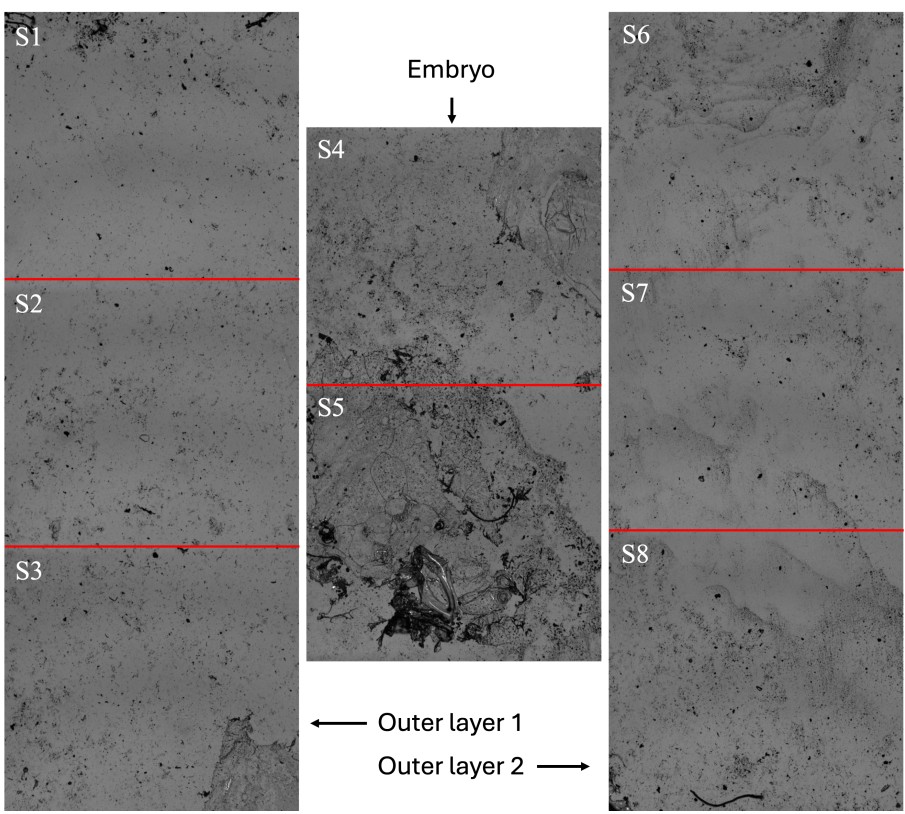

**Figure 2.** 2-D cross-section along an axis in the equatorial plane at 108x magnification for sample V-7 collected on 8 February 2018. Sectors (S) 4 and 5 indicate the embryo, while S1-S3, S6-S8 represent the outer layers.



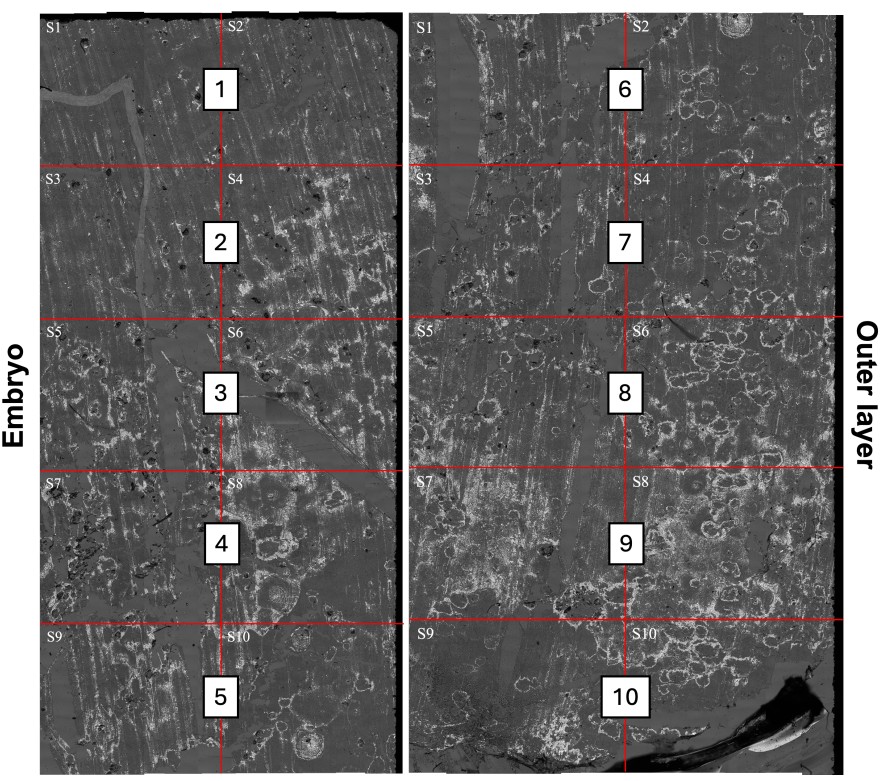

**Figure 3.** As in Figure 2, but for NG-1 collected on 13-14 December 2018. Sectors are grouped by layers relative to the embryo (layers 1-5) and outer layer (6-10).



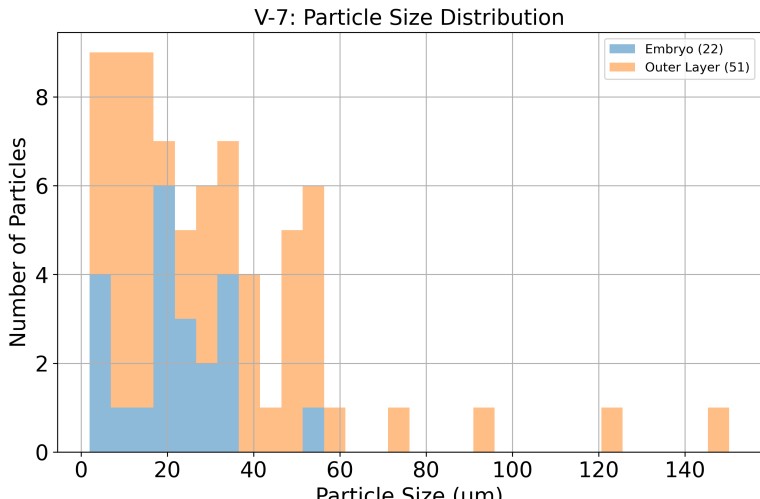

**Figure 4.** Particle size distribution in $\mu$m through the V-7 cross-section is shown in Figure 2, where the particle size distribution in the hailstone embryo region is represented by blue pastel bars, and the particle size distribution in the outer layers of the hailstone is shown by orange pastel bars.



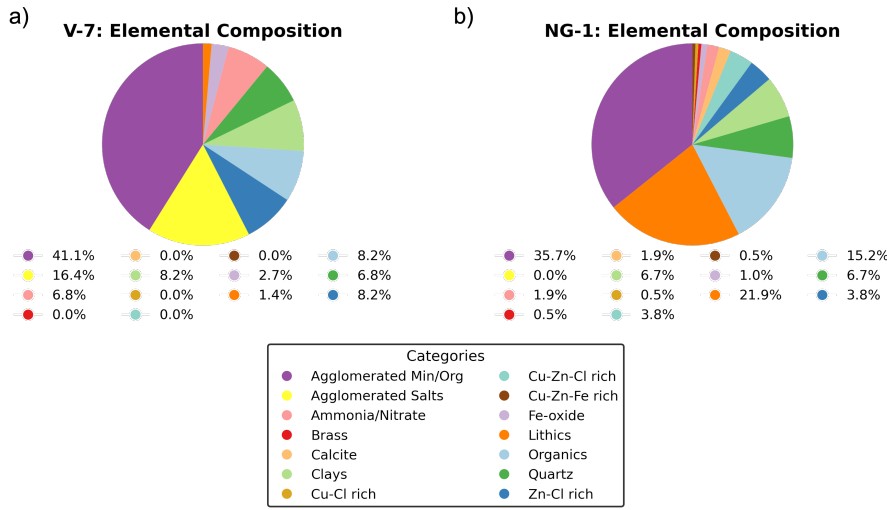

**Figure 5.** Elemental composition distributions for particles selected within the 2-D cross sections for a) V-7 and b) NG-1.

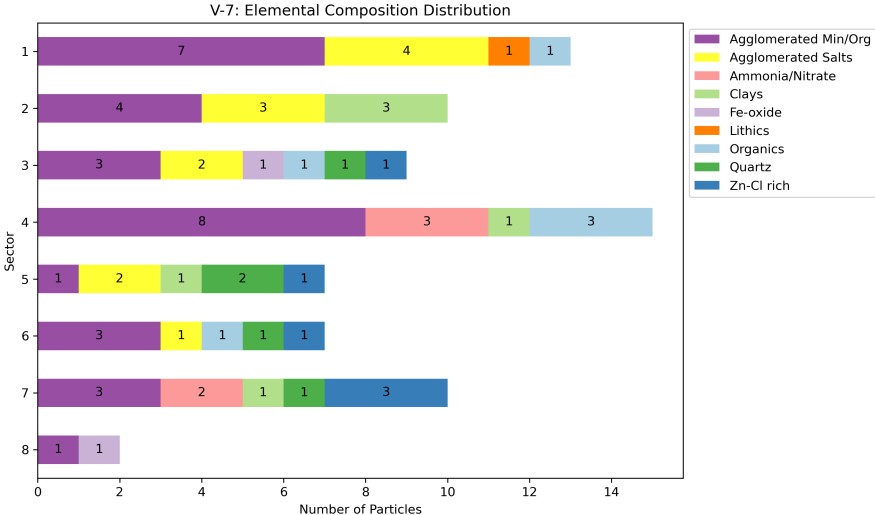

**Figure 6.** Elemental composition distribution for particles identified in the V-7 cross-section shown in Figure 2, where Sectors 5 and 6 represent the embryo region.



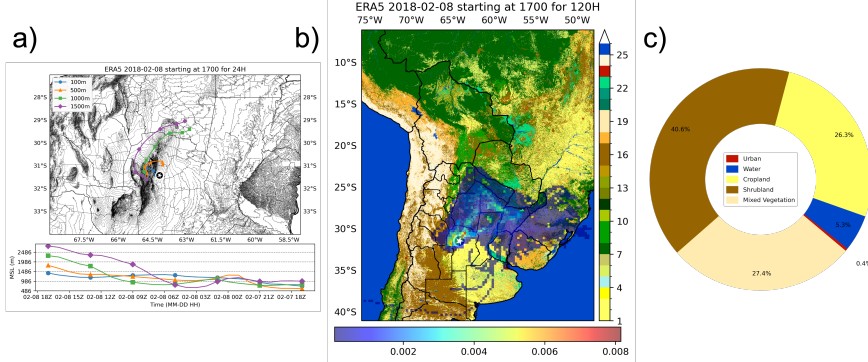

**Figure 7.** HYSPLIT analysis for 8 February 2018. (a) (top) shows 24-hour HYSPLIT back trajectories color-coded by height (in meters), with Córdoba City marked by a white star. The bottom of (a) displays terrain analysis in meters above sea level for the same time periods as the back trajectories. (b) shows residence time coefficients calculated for 5-day back trajectories overlaid on the C3S Land Cover map with Córdoba City marked by a white star. The land cover colors correspond to a subset of categories shown in Figure 1, specifically: 1/2/3-Cropland rain-fed, 4-Cropland irrigated, 5-Mosaic cropland, 6-Mosaic natural vegetation, 7-Evergreen broadleaved, 8/9/10-Deciduous broadleaved, 11-Evergreen needleleaved, 12-Mixed trees, 13-Mosaic tree/shrub, 14-Mosaic herbaceous, 15/16-Shrubland, 17-Grassland, 18-Sparse vegetation, 19-Sparse herbaceous, 20-Fresh water flooded tree cover, 21- Saline water flooded tree cover, 22-Flooded shrub/herbaceous cover, 23-Urban, 24-Bare areas, 25-Water, 26-Snow and ice. (c) displays the predominant land uses within the residence time coefficient pixels observed in b). Similar land uses were grouped into the following categories: Urban (23), Water (25,26), Cropland(1,2,3,4,5), Shrubland(15,16,17), Mixed Vegetation (6,7,8,9,10,11,12,13,14,18,19). Flooded Vegetation (20,21,22) and bare areas (24) were not included.

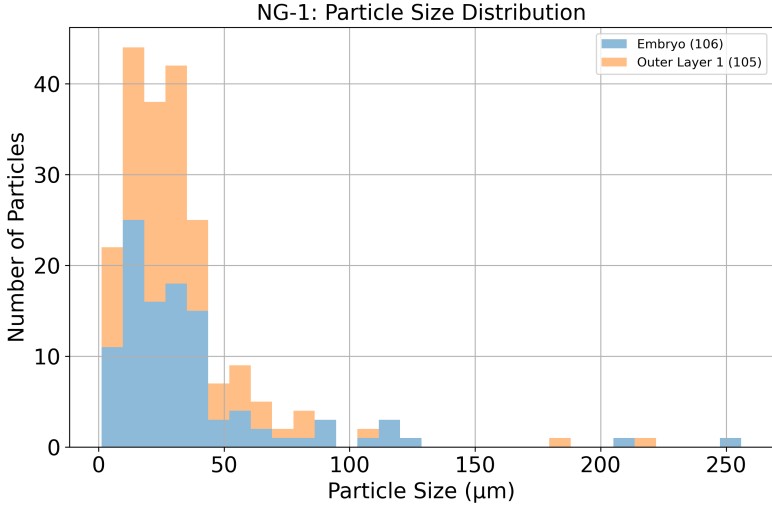

**Figure 8.** Particle size distribution in $\mu$m through the NG-1 cross-section is shown in Figure 3, where the particle size distribution in the hailstone embryo region is represented by blue pastel bars, and the particle size distribution in the outer layers of the hailstone is shown by orange pastel bars.

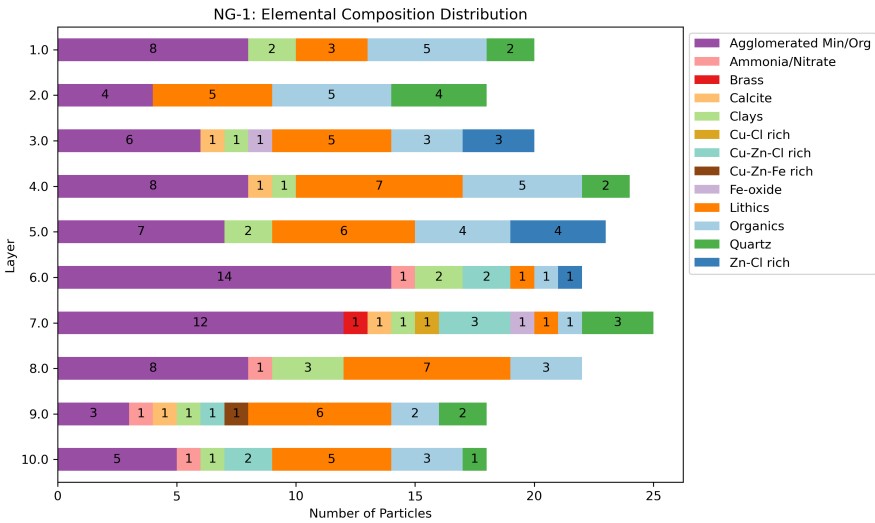

**Figure 9.** Elemental composition distribution for particles identified in the NG-1 cross-section shown in Figure 3 where Layers 1-5 represent the embryo region.

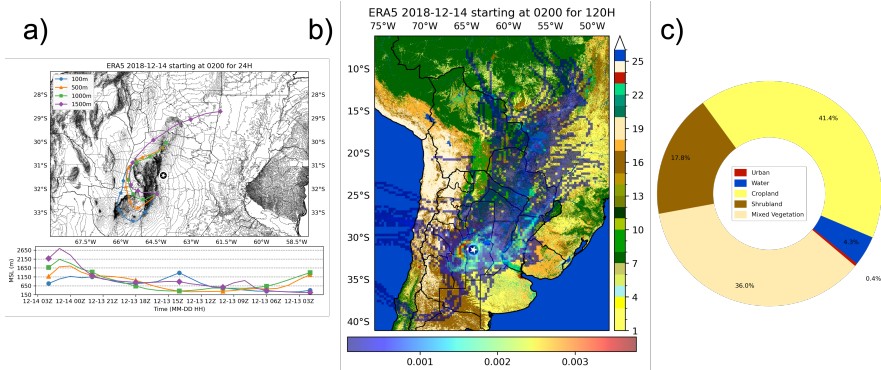

**Figure 10.** HYSPLIT analysis for 13-14 December 2018. (a) (top) shows 24-hour HYSPLIT back trajectories color-coded by height (in meters), with Córdoba City marked by a white star. The bottom of (a) displays terrain analysis in meters above sea level for the same time periods as the back trajectories. (b) shows residence time coefficients calculated for 5-day back trajectories overlaid on the C3S Land Cover map with Córdoba City marked by a white star. The land cover colors correspond to a subset of categories shown in Figure 1, specifically: 1/2/3-Cropland rain-fed, 4-Cropland irrigated, 5-Mosaic cropland, 6-Mosaic natural vegetation, 7-Evergreen broadleaved, 8/9/10-Deciduous broadleaved, 11-Evergreen needleleaved, 12-Mixed trees, 13-Mosaic tree/shrub, 14-Mosaic herbaceous, 15/16-Shrubland, 17-Grassland, 18-Sparse vegetation, 19-Sparse herbaceous, 20-Fresh water flooded tree cover, 21- Saline water flooded tree cover, 22-Flooded shrub/herbaceous cover, 23-Urban, 24-Bare areas, 25-Water, 26-Snow and ice. (c) displays the predominant land uses within the residence time coefficient pixels observed in b). Similar land uses were grouped into the following categories: Urban (23), Water (25,26), Cropland(1,2,3,4,5), Shrubland(15,16,17), Mixed Vegetation (6,7,8,9,10,11,12,13,14,18,19). Flooded Vegetation (20,21,22) and bare areas (24) were not included.