# Peer review of "Physical-chemical properties of particles in hailstones from Central Argentina"

_EGUsphere, 2024_

## Author Response (AR1)

**Reviewer 1**

Overview:

In this manuscript the authors describe the particles associated with 2 hail particles using confocal microscopy for imaging and SEM for imaging along with EDS in the SEM for elemental analysis. The technique used to collect and preserve the particles specially was thought out but has some significant issues when trying to quantify the results with a thick layer of polymer and then coated with gold. Drawing a lot of the conclusions from small number of particles is difficult. Cut multiple layers to get more statistics on the exterior particles. There is a lot of potentially interesting data, but needs to be analyzed better/concisely. The major conclusion seems to be that the embryo is not different than the outer shell particles? Also the particles in each are similar? This needs to be shown better by the results that are to spread out. Analysis is concerning about the points the paper tries to make from very few number of particles. The EDS assumptions as described below have some issues that would cause erroneous results. Overall I think there is definitely some interesting data here that needs further analysis and writeup to be more convincing. I would suggest to possibly reconsider this paper with very major changes and revisions.

Thank you for providing a thorough and constructive review. We understand your overall concerns and will address them in the individual comments below.

General Comments:

The term 'non-soluble' is used throughout. This would be true when they talk about melted samples that are analyze whereas here the ice is sublimed and that would leave soluble particles as well as insoluble particles.

This is a good point in recognizing the complexity of solubility with the potential of more than water molecules to sublimate and the need to be careful with our terminology. To address this important point, we have modified the text to acknowledge that our sublimation method could preserve both soluble and insoluble particles, unlike melting methods that would dissolve soluble particles. We clarified this distinction when first introducing our methodology in lines 126-127 as follows: "Unlike traditional melting methods, which would dissolve soluble particles, our sublimation approach potentially preserves both soluble and non-soluble particles, though this study focuses on characterizing non-soluble particles. Through this method, we can observe both the non-soluble particles and deposits of compounds that did not sublimate."

Better comparison of core vs. shell. This gets lost with some of the singling out of smaller regions (e.g. Figure 6 could be embryo vs. outer shell to get better statistics) otherwise just noise that you cannot make conclusions from.

We appreciate your suggestion to improve the comparison between the core and shell. We have revised Figures 7 and 10 to present the data as a comparison between the embryo and the outer layers, which provides clearer distinctions and enhances the interpretability of the elemental composition distribution.

Need to include SEM imagery of particle. Talked about aggregated particles a bit with no supporting images. You should be able to get better size information than with confocal.

Thank you for this suggestion. We have provided example SEM images of particles to address this point with its corresponding EDS spectra for your next point. In particular, we have shown an example agglomerated particle, highlighting its composition of many elements and its visually irregular shape, a clear example of our Fe-oxide category, and our calcite category; all of which were found in both the embryo and outer layers and in both samples highlighted in this paper. These example particle images have been added to the paper as Figure 4 with the corresponding text description in lines 203-214.

There needs to be the EDS shown in reference to particles. This could be some nice supplementary information. Additionally please show example EDS spectra for each particle type.

We appreciate the reviewer's suggestion regarding EDS spectra presentation. To address this point, we have added a new figure (Figure 4) that showcases three representative examples of our particle analysis workflow, including CLSM images, SEM micrographs, and their corresponding EDS spectra. These examples were carefully selected to demonstrate:

An agglomerated mineral/organic particle (the most common category in both hailstones)

A Fe-rich organic particle (showing clear elemental signatures)

A salt particle (demonstrating distinct Na and Cl peaks characteristic of halite)

This approach allows readers to understand our classification methodology while avoiding redundancy, as many particle types share similar elemental signatures with varying relative abundances. The complete EDS dataset is available in our public repository for researchers interested in examining the full spectral collection.

Bernal Ayala, A. C., Rowe, A., Arena, L. E., and Nachlas, W. O.: Individual Particle Dataset: Physical-chemical Properties of Non-Soluble Particles in a Hailstone Collected in Argentina, https://doi.org/10.5281/zenodo.10455803, 2024a.

EDS issues are mentioned, but subsequently ignored especially in relationship to carbon and oxygen going through 10-20nm of Au and 1micron plus of formvar. See simple Montecarlo simulation at 15 kV below for EDS and imaging issues with some conclusions using this approach. 10 nm Au, 1um Formvar, 10 micron sample (used CaCO3), substrate

Our understanding from this comment was that there was supposed to be a figure provided with this review showing the output of a Montecarlo simulation, but unfortunately the figure was not included so we cannot directly address this comment with that output in mind. However, we realize we could better clarify this point in our paper by referring to the Montecarlo simulation and EDS output from Formvar and glass we provided in a previous paper (Bernal Ayala et al. 2024; Figures 8, 14, and 15; copied below for your convenience). Results from our Montecarlo simulation (Figure 8 of that paper) indicate that for a small biotite particle with a 7 nm layer of Formvar, some x-rays are emitted from the glass substrate for particles smaller than 1.5 microns at 15 kV that is minimized using the 8 kV accelerating voltage. This showed minimal impact of the soda-lime glass for particles down to 1 microns but then we used a single-point analysis for our EDS spectra to further reduce any spectral contamination. So while the single point analysis was more labor intensive, it increased our ability to reduce noise.

Figures 14 and 15 in that paper further highlighted our approach by sampling the EDS spectra of the glass then removing it from our particle+glass spectra to emphasize the elements that would go into our particle type classification. While this approach showed a minimal contribution of carbon from the glass to our interpretation of the spectra, we chose to classify our particles based on element identification rather than quantifying with weight percentage values. This approach ensures the reliability of our results despite the potential issues with carbon and oxygen signals. We recognize that this revised manuscript should include more of a mention of the outcome of the work in our 2024 methods paper toward explaining these potential limitations of the glass substrate and therefore made the following changes in our methods sections of the revised manuscript (lines 169-172):

[Figure]

[from Bernal Ayala et al. 2024] Figure 8. Activation volume producing Fe X-rays within the soda–lime glass for 8 kV (a) and 15 kV (b) accelerating voltage from the Monte Carlo simulations. Lines indicate the pathways of individual electrons backscattered from within the sample (blue) and from the sample surface (red) as a function of the particle size (in nm).

[Figure]

[From Bernal Ayala et al. 2024] Figure 14. Mathematical operations on the spectrum using DTSAI were done to analyze the EDS spectrum data and differentiate a particle's spectrum from the glass substrate. Panel (a) depicts the spectrum of the particle presented in Fig. 7b. Panel (b) illustrates the spectrum of the glass. Panel (c) demonstrates the outcome of the subtraction, revealing the presence of elements C, Na, Cl, and K in the particle. The first peak, closest to 0 keV, is caused by noise from the EDS detector and can be ignored when analyzing the elemental composition of a particle, as it does not represent any actual element. The second spectral peak corresponds to gold (2.12 keV) and was excluded from the analysis since it was used for coating the sample.

[Figure]

[From Bernal Ayala et al. 2024] Figure 15. Panels (a), (c), and (e) display a single-point spectral analysis of a C-based particle with Formvar (a), a clear area with Formvar (c),and a clear area without Formvar (e). Each panel in the figure to the right (b, d, f) is the corresponding panel providing a closer view of the C peak. Due to Formvar's thickness and the results in this figure, the Formvar layer's impact on the C peak is considered negligible. The first peak, closest to 0 keV, is caused by noise from the EDS detector and can be ignored when analyzing the elemental composition of a particle, as it does not represent any actual element. The second spectral peak corresponds to gold (2.12 keV) and was excluded from the analysis since it was used for coating the sample.

**Bernal Ayala, A. C., Rowe, A. K., Arena, L. E., Nachlas, W. O., and Asar, M. L.: Exploring non-soluble particles in hailstones through innovative confocal laser and scanning electron microscopy techniques, Atmos. Meas. Tech., 17, 5561–5579, https://doi.org/10.5194/amt-17-5561-2024, 2024**

Specific Comments:

41: melting only looks at non-soluble particles, it removes soluble particles

Good clarification point. We have made the following edit to make sure it's clear that melting removes the ability to discern non-soluble particle information with respect to its spatial distribution within the stone.:
"all required melting the hailstones, removing information on non-soluble particle size distribution or composition with respect to the hailstone embryo, neglecting soluble particles." (Lines 62-64)

89: How collected and transported?

The hailstone sample was transferred and kept in a nylon bag in a thermally reinforced cooler in which the temperature was controlled at −15 °C. Each sample was designated an alphanumeric identifier that included the name of the person who collected the sample. Once it reaches the laboratory, the hailstone surface is brushed (to remove contamination) and transferred to a cold chamber with a controlled temperature of −15 °C. This is referred to in the published methods paper Bernal Ayala et al. 2024. We've added this reference in this context in lines 114-115

Bernal Ayala, A. C., Rowe, A. K., Arena, L. E., Nachlas, W. O., and Asar, M. L.: Exploring non-soluble particles in hailstones through innovative confocal laser and scanning electron microscopy techniques, Atmos. Meas. Tech., 17, 5561–5579, https://doi.org/10.5194/amt-17-5561-2024, 2024

95: how close to the center? How did you know it was the embryo and directly grown from that point?

For these two cases, the embryo is identified as the original nucleus through its uniform crystallographic structure under polarized light, contrasting with the radially aligned outer layers (Grenier and Sadok Zair, 1983; Takahashi, 1987). The core's small, equiaxed ice grains share nearly identical c-axis orientations, indicating it formed from a single frozen droplet or graupel particle (Takahashi, 1987). This uniformity distinguishes it from the outer shells, where elongated grains align radially outward due to wet accretion (water freezing onto the surface; Takahashi, 1987). The radial alignment of c-axes in these layers points to concentric growth around the embryo, while the embryo's central position and structural

simplicity confirm it as the starting point (Soderholm and Kumjian, 2023). The transition from a single-oriented core to layered shells reflects the hailstone's formation sequence, where the embryo acts as the nucleation site for subsequent accretion (Takahashi, 1987; Soderholm and Kumjian, 2023). Additionally, in porous hailstones, the distribution and size of air bubbles differ significantly between the central zone and the external lobes, providing another distinguishing characteristic between the embryo and outer layers (Lubart and Levi 1984, Levi et al. 1989, Levi et al. 1991). These have been added in lines 134-142.

104-108: Figures 2 and 3 are very confusing with different ways of designating the core/shell and how the sample is analyze. It would be helpful to keep these consistent (with the S# I was trying to find the supplemental information) and also show how they are related to each other distance wise since it is unclear in the figure. It would be great to stitch the entire confocal images together of the entire sample and maybe put that in supplemental information and pull images from that for the main document.

We understand there is confusion here, given the differences between the hailstones. We tried different layouts of these figures to see an appropriate balance between the orientation of the figures and their sizes to see the detail. That said, a scale will go a long way to help clarify our point, and therefore, it has been added to these figures to emphasize the relative sizes. We also realize the confusion we created by referring to sections of Figure 2 by S1, S2, etc., in the text in that nomenclature would typically be reserved for referring to supplemental figures. Therefore, in the revised manuscript,  Sector is spelled out when referring to each sector in the text.

131: Single point EDS is not sufficient to get particles information about micron size particles, it would be better to average over most of the particle area to get an overall particle idea if they are agglomorates.

We understand that averaging over the entire particle area could provide more information about the particle composition. However, we chose a single-point approach to minimize the risk of the electron beam hitting the glass slide, which could introduce erroneous peaks as described in our Bernal Ayala et al. 2024 methods paper. For validation of this approach, we conducted multiple point measurements on sufficiently large particles where we could safely avoid both edge effects and glass slide interference. As demonstrated in the figure below, we show two representative particles that were classified in different categories: an agglomerated salt particle (V-7 S1 PA3, left column) and an agglomerated mineral/organic particle (V-7 S2 PA7, right column).

For the agglomerated salt particle, three separate point measurements (Spectrum 1-3) show consistent elemental signatures with nearly identical relative peak intensities for Na and Cl. When averaging the weight percentages across all three spectra, the particle still clearly falls within our classification criteria for agglomerated salts, with strong Na-Cl signals dominating the composition. Similarly, for the agglomerated mineral/organic particle,

four-point measurements (Spectrum 1-4) demonstrate consistency in the elemental composition, with Si being the dominant peak and consistent minor contributions from Na, O, and C. The averaged weight percentages from these four spectra maintain the characteristic signature of agglomerated mineral/organic particles, with significant Si content and the presence of other elements typical of mineral and organic matter.

The SEM images further support these classifications, with the agglomerated salt particle showing an irregular, crystalline structure characteristic of salt aggregates, while the agglomerated mineral/organic particle displays a more compact, heterogeneous surface typical of mineral-organic mixtures. These results provide confidence in our single-point measurement approach, showing that for particles in our samples, multiple measurements did not reveal significant compositional heterogeneity that would have been missed by single-point analysis, and the averaged compositions consistently support our particle classifications. This approach ensured the reliability of our results while minimizing potential interference from the substrate.

[Figure]

136: "unique feature… particles with location" This needs to be more pronounced in the paper. You have some size information and it would be good to talk about size and composition versus distance from the embryo on the paper

We have addressed this by revising Table 1 to explicitly show particle size ranges separated by location (embryo vs. outer layers) for each compositional category. The revised Table 1 reveals several important spatial relationships. For instance, in sample NG-1, Agglomerated Min/Org particles show size distributions between the embryo (4-77 μm) and outer layers

(9-109 μm). Together, with Figures 4 and 6, we observe that certain particle types are location-specific; Cu-Cl rich and Cu-Zn-Cl rich particles are exclusively found in NG-1's outer layers, while Agglomerated Salts are only present in V-7. Even for particle types present throughout the hailstone, such as Quartz in V-7, we observe different size ranges between the embryo (16-22 μm) and outer layers (38-55 μm). These details corresponding to revised table 1 are provided in Section 3.1.1 (lines 252-298) and 3.2.1 (lines 339-369).

This tabular format complements our existing figures: Figure 4, which presents the overall size distribution analysis, and Figure 6, which shows the compositional comparison across locations. Together, these three presentations (Table 1, Figure 4, and Figure 6) provide a detailed characterization of how particle properties vary with location within the hailstone structure, addressing the spatial relationships the reviewer has highlighted.

146: Deconvolution will not be good especially for carbon and provide very unreliable results since you don't know exact thickness of formvar, Au, or particle. Plus heterogeneity will cause issues. Carbon coating would be better than Au to have less overlap of peaks and absorption of signal.

Our approach focuses on using EDS for element identification rather than quantifying exact abundances, particularly for carbon. The challenges mentioned, such as the variability in Formvar thickness, reinforce our decision to focus on identifying the presence of elements rather than their precise concentrations. While the exact thickness of the Formvar coating may vary, this primarily affects carbon quantification, which is not the focus of our analysis. Further, we are confident that the small carbon signal from the Formvar can be effectively ignored for our purposes of element identification, as shown in Bernal Ayala et al.2024 and described in a previous response. We have revised section 2.1.2 in our manuscript (lines 185-188) to better integrate this discussion of EDS limitations and how they lead to our specific approach for particle classification, now also mentioning the formvar thickness variability.

The choice of Au and Au-Pd coatings was made to enhance high-resolution microscopy imaging. Gold is generally preferred as a coating material because it provides enhanced image quality due to its higher secondary electron yield, resulting in better SEM images with improved contrast and detail (Goldstein et al., 2017). Also, gold's high conductivity effectively mitigates sample charging, especially for non-conductive samples, leading to clearer and more stable images (Reimer, 1998). It is particularly beneficial for imaging at low to medium magnifications, making it suitable for a wide range of applications (Egerton, 2005). Additionally, the typical gold coating grain size is 8–12 nm, which is often smaller than the resolution limit of many SEM instruments, especially those with thermionic emitters (Goodhew et al., 2001). While we acknowledge the potential for peak overlap, this coating was chosen to achieve the imaging quality required for our analysis.

**Reimer, L. (1998). Scanning Electron Microscopy: Physics of Image Formation and Microanalysis (2nd ed.). Springer Science & Business Media.**

**Egerton, R. F. (2005). Physical Principles of Electron Microscopy: An Introduction to TEM, SEM, and AEM. Springer Science & Business Media.**

**Goodhew, P. J., Humphreys, J., & Beanland, R. (2001). Electron Microscopy and Analysis (3rd ed.). Taylor & Francis.**

153-157: Extraneous, you are doing the analysis not the instrument and need to ID them and understand not say what the problems are of possibilities.

In rereading this section, we understand your point. The purpose of including this list of limitations here was to justify using presence and relative abundances rather than exact quantities for our particle classification method. By highlighting the limitations of EDS in measuring elemental abundances, we aim to justify our methodological approach. We feel much of this is needed, although agree that this particular section is not needed and so we have a revised version that trims this section down to get to the key point here about our focus on identifying the presence of elements due to these limitations.

158: 'activation' should be 'interaction'

We acknowledge your recommendation. While the distinction between the terms is technically valid, both are contextually appropriate. However, we decided to keep the same verbiage consistent with our precious publication Bernal Ayala et al, 2024.

160: 'we did not place high confidence' – just say what you did.

We understand this point and in the context of your other comments, we've rewritten this section to streamline our point.

207: 1.9 um – is this area equivalent diameter? You did EDS on these particles, can you measure more accurately? It is concerning with a thickness of 1micron+ of formvar getting much BSE signal out of the sample (see a Montecarlo simulation).

Thank you for this important technical question. Our particle size measurements were conducted using SEM imaging, where we measured the longest axis of each particle. The uncertainty in these measurements is ±1 μm, which accounts for both the imaging resolution and the potential effects of the Formvar coating. This measurement approach was applied consistently across all analyzed particles, regardless of size.

Regarding the EDS analysis and your concern about BSE signal attenuation through the Formvar coating: Our sample preparation method produces a thin Formvar film using a 1% solution. This thin coating allows for reliable BSE signal collection and EDS analysis, as demonstrated by:

1. Clear topographic contrast in our BSE images
2. Consistent elemental signatures across multiple measurements of the same particle type
3. Well-defined peaks in our EDS spectra, even for smaller particles

While we acknowledge that Monte Carlo simulations would provide additional validation of beam-sample interactions through the Formvar coating, our consistent results across hundreds of analyzed particles support the reliability of our measurement approach.

211: Hard to make argument of size since the embryo particles are only analyzing 22 particles and the particles in the shell are mostly similarly sized. Larger ones could have just been cut down or missed easily

This is a good point in how we can be more careful in our wording of results to not overgeneralize. In this particular line, the main point here was how large the particles were in both the outer layer and embryo in particular. To address your comment and make thise point clearer, we've revised lines 259-261 to say "While the comparison between the embryo and outer layers in V-7 suggests that the embryo contained larger particles than the outer layers, a key point here is that both regions contained particles exceeding 100 μm."

215/Figure 5: 15 different classifications for only small number of particles makes this not significantly relevant. Cut down on the number of classifications since there are multiples with 0%

We argue here that the distinctions between categories are important and while the lack of a category in our particle samples does not confirm it was not present at all in the hailstone, these distinct categories help us highlight what was present and associated differences between the stones even if only one of particles makes that point. We want to keep the 0% in the figures as it makes it more straightforward to compare the two samples, particularly re-using the associated legend. But we do recognize that we don't want to be misleading with the small percentages and therefore have modified our wording in the text associated with these figures and in our discussion to this effect, notating how many particles encompass each percentage. These have been discussed in Sections 2.1.2, 3.1.1 and 3.2.1.

219: range from 2-75 microns. This is a few particles, the range is basically meaningless since you are basically saying it is the few particle analyzed.

We acknowledge the reviewer's observation regarding the reporting of particle size ranges. To address this concern, we have updated Table 1 to provide more meaningful size information based on the number of particles in each category. For categories containing three or more particles, we report the size range (minimum to maximum). For categories with only one or two particles, we list the individual particle sizes separated by commas rather than presenting them as a range. This approach provides a more accurate and

transparent representation of our particle size data, avoiding potentially misleading ranges when the sample size is very small. This revised presentation better reflects the actual distribution of particle sizes within each category while acknowledging the limitations of small sample sizes in some categories.

220: 'Agglomorated salts' these are soluble particles. Make sure to change around the solubility

While salts are typically soluble, our introduction discusses how they can serve as ice nucleating particles (INPs) and potentially remain intact. This is possible when they are compacted with other particles or coated (Patnaude et al., 2021). These conditions can prevent dissolution and allow salts to maintain their structure, supporting their role in our analysis.

**Patnaude, R. J., Perkins, R. J., Kreidenweis, S. M., and DeMott, P. J.: Is Ice Formation by Sea Spray Particles at Cirrus Temperatures Controlled by Crystalline Salts?, ACS Earth Space Chem., 5, 2196–2211, https://doi.org/10.1021/acsearthspacechem.1c00228, 2021.**

223: BSE imagery did not show cubic crystal – please show some SEM images of these? This could be from a brine solution that sublimed and did not form a nice single cubic particle.

We adjusted the reference to cubic crystal structure and revised the text to describe the observed morphology more accurately. The relevant text now reads:

"In contrast, the SEM image displays an irregular shape with uneven edges and lacks the symmetrical, straightedge characteristics typically associated with cubic morphology, typically associated with halites." (Lines 213-214)

This revised description better reflects what we observe in Figure 4c, where the salt particle shows an irregular morphology. We maintain the particle's classification as a salt based on its strong Na-Cl EDS signature while acknowledging that its morphology differs from the ideal halite crystal structure.

228: Cl could be problematic since you see a lot of NaCl above that is not crystalline single particles.

It's true that Na and Cl could form secondary crystals if they were dissolved in the ice and then precipitated during sublimation. However, our observations suggest that these are not secondary particles. We only detected NaCl in specific layers of the hailstone, not throughout the entire sample. If NaCl had precipitated from a solution during sublimation,

we would expect it to be present in all spectra across the sample, which was not the case. This selective presence supports our interpretation that the NaCl particles were originally captured by the ice.

245: 'no detectable carbon' there is a 1um layer of formvar. No way you can say there is no carbon present.

The 1 μm layer of Formvar does not produce carbon X-rays at 15kV, which is why we state "no detectable carbon" in the context of our EDS analysis. We interpret all carbon present in the spectrum to be within the particle itself, not from the Formvar coating. As detailed in Bernal Ayala et al. (2024), our analysis confirmed that the contribution of carbon from the Formvar is minimal. We compared potential carbon signals in clear glass sections with and without Formvar, and the results showed negligible impact from the Formvar layer. We will ensure this clarification is explicitly stated in the manuscript.

The text has been change to:
"…no detectable carbon from the Formvar." Lines 290-291

The carbon detected in the spectrum is interpreted to be within the particle itself. As detailed in Bernal Ayala et al. (2024), the contribution of carbon from the Formvar is minimal, with negligible impact on the carbon signal.

293: 35+/-34 microns is not anything useful. Maybe range is, but hard to make conclusions like this with such a small data set.

I have removed the standard deviation from the results, and only included the average particle size. Table 1 has also been updated to make this link clearer, mentioned in text how many particles make up the relative percentages for the composition pie charts.

346: 'size, composition, and distribution' it would be great to actually compare all of these. They are spread out and composition vs. size is not shown.

We have revised Table 1 to better illustrate the relationships between particle size, composition, and spatial distribution within the hailstones. The table now explicitly separates particle size ranges between embryo and outer layers for both V-7 and NG-1 samples, revealing several key spatial patterns that were not immediately apparent in the previous version.

This enhanced presentation demonstrates important relationships such as the variable size distributions of Agglomerated Min/Org particles between locations (V-7: 6-75 μm in embryo, 2-55 μm in outer layers; NG-1: 4-77 μm in embryo, 9-109 μm in outer layers). It also highlights location-specific particle types, such as Cu-Cl rich and Cu-Zn-Cl rich particles found exclusively in NG-1's outer layers, and Clays found only in V-7. Furthermore, the table reveals size range variations for particles present in both locations, as exemplified by

Organics in NG-1 showing broader size ranges (9-256 µm in embryo, 13-216 µm in outer layers) compared to V-7 (22-33 µm in embryo, 10-23 µm in outer layers).

We believe this revised presentation provides readers with a clearer understanding of how particle characteristics vary both by location within individual hailstones and between different hailstone samples, thereby strengthening the manuscript's analysis of particle distribution patterns. Thank you for this suggestion.

354: 'carbon (76%) and oxygen (22%)' – there is a 100 micron layer of C/O on top of this. It is impossible to try and quantify the C:O ratios of this with how the procedure was performed.

Thank you for your comment. We would like to clarify that the Formvar coating applied in our procedure is approximately 1 micron thick, not 100 microns. While there may be some variability in the thickness of the Formvar layer across the sample, it does not approach the magnitude of 100 microns. This thin layer is unlikely to significantly affect the quantification of carbon and oxygen ratios in the particles themselves. As previously mentioned, the carbon detected in the EDS spectra is interpreted to originate from the particles, not the Formvar coating. Our methodology ensures that the contribution of carbon from the Formvar is minimal, as detailed in Bernal Ayala et al. (2024), where we demonstrated that the impact of the Formvar layer on the carbon signal is negligible.

357-58: 'particle agglomeration played a significant role' – please show imagery of particles. You mention lots of morphology throughout the paper, but there are no SEM images of the particles to show any of these conclusions.

Our new Figure 4 in the revised manuscript shows an example of an agglomerated min/org particle, including its SEM imagery and EDS spectra.

375: the Cu/Cl can just be agglomorates, hard to tell without higher resolution data.

While it is possible that the Cu-Cl presence in the NG-1 sample could be due to agglomerates, this aligns with a key focus of our study, which is the significance of particle agglomeration. We acknowledge that higher-resolution data could provide more definitive insights. However, it is important to note that with a larger sample size and replication of this method, literature suggests that similar findings could indeed link these particles to specific sources, such as agrochemicals like copper chloride. Even if the particles are coated, the presence of such coatings is significant and was successfully categorized using the analytical scheme discussed in this paper. This categorization provides valuable insights into the potential sources which can be valuable when studying the environmental implications of these particles.

**Reviewer 2**

This paper by Bernal et al., provides a comprehensive analysis of non-soluble particles trapped within hailstones collected in Central Argentina, offering novel insights into their physical and chemical properties, sources, and implications for hail formation processes. By employing innovative methodologies such as Confocal Laser and Scanning Electron Microscopy combined with Energy-Dispersive Spectroscopy, the authors effectively preserved and analyzed the in-situ characteristics of particles, marking a significant advancement over traditional methods. The study is well-grounded in its regional focus, exploring the link between local land-use types and particle sources, while also addressing broader implications for understanding hail formation under varying storm modes. The manuscript is technically robust, methodologically detailed, and contributes valuable findings to the fields of atmospheric science and aerosol research. However, to maximize the study's impact, the manuscript would benefit from a more concise presentation of results, a clearer articulation of its global relevance, and streamlined technical details to enhance accessibility for a broader audience. Overall, the study makes a substantial contribution to the understanding of hail microphysics and its environmental drivers, warranting publication with minor revisions.

Thank you for constructive comments to help us clarify our results and their relevance. We have addressed your specific comments below.

Here are my suggestions for improvement:

Suggestions for Improvement:

1. Highlight the broader significance of understanding hail microphysics in the context of extreme weather forecasting, climate modeling, and hydrological cycles. This will help position the study as not just regionally significant but also globally relevant.

This has now been addressed in lines 20-33.

2. Ensure consistent usage of terminology, particularly between "non-soluble particles" and "insoluble particles," to avoid confusion.

The text has been revised, and all "insoluble particles" instances have been updated to be consistent with our choice of using "non-soluble" particle terminology.

3. Provide a rationale for the selection of key parameters, such as the particle size threshold (e.g., 1 μm) and the use of specific coating materials (gold and gold-palladium). This would help readers understand how these choices were optimized for the study.

Thank you for this comment. We recognize that we relied heavily on the reader referring back to our Bernal Ayala et al. 2024 methods paper that described all of these details as we did not want too much repetition of the detailed methods choices here in this paper to focus more on results of this new method. We'll clarify these choices for you here with the acknowledgement that we feel it's too much detail to include in this paper when much of it is provided in our recently published methods paper.

When attempting to analyze a non-flat particle covered by a 1-micron layer of polyvinyl formal on a glass substrate, several factors converge to create practical resolution limitations of approximately 1 micron, despite the theoretical capabilities of both confocal microscopy and energy-dispersive X-ray spectroscopy. The polyvinyl formal coating creates a significant refractive index mismatch boundary that directly impacts imaging performance. According to research on confocal microscopy, refractive index mismatches between the mounting medium, sample, and the glass-oil immersion system result in spherical aberrations that substantially degrade axial resolution when imaging beyond the coverslip surface. These aberrations cause point-spread function broadening, decreased signal intensity with depth, and focal shift distortions along the z-axis [1].

The non-flat nature of the particle further exacerbates these challenges by creating variable optical path lengths through the polyvinyl formal layer. This variability introduces additional aberrations beyond simple spherical aberration, as light rays encounter inconsistent thicknesses when traversing the sample. Studies have demonstrated that mounting media with refractive indices precisely matching the glass-oil immersion system (approximately 1.518) maintain constant axial resolution throughout depth, while mismatched media show significant degradation [1].

For EDX analysis, the electron beam must penetrate the entire 1-micron polyvinyl formal layer before reaching the particle of interest, which increases the interaction volume and produces a larger excitation region. This expanded interaction volume directly limits spatial resolution and increases the background signal from the polyvinyl formal layer itself. The irregular surface topography also creates varying incidence angles for the illumination and collection pathways in confocal microscopy and similar challenges for electron beam interactions in EDX spectroscopy, effectively establishing a practical resolution limit of approximately 1 micron for this specific sample configuration.

Au and Au-Pd coatings were chosen to enhance high-resolution microscopy imaging. Gold is generally preferred as a coating material because it provides enhanced image quality due to its higher secondary electron yield, resulting in better SEM images with improved contrast and detail (Goldstein et al., 2017). Also, gold's high conductivity effectively mitigates sample charging, especially for non-conductive samples, leading to clearer and more stable images. It is particularly beneficial for imaging at low to medium magnifications, making it suitable for various applications (Reimer, 1998). Additionally, the typical gold coating grain

size is 8-12 nm, often smaller than the resolution limit of many SEM instruments, especially those with thermionic emitters (Goodhew et al., 2001). While we acknowledge the potential for peak overlap, this coating was chosen to achieve the imaging quality required for our analysis.

[1] https://pmc.ncbi.nlm.nih.gov/articles/PMC4379090/

**Reimer, L. (1998). Scanning Electron Microscopy: Physics of Image Formation and Microanalysis (2nd ed.). Springer Science & Business Media.**

**Egerton, R. F. (2005). Physical Principles of Electron Microscopy: An Introduction to TEM, SEM, and AEM. Springer Science & Business Media.**

**Goodhew, P. J., Humphreys, J., & Beanland, R. (2001). Electron Microscopy and Analysis (3rd ed.). Taylor & Francis.**

4. Include scale bars on Figures 2 and 3 to provide accurate spatial context for the microscopy images. Ensure the scale bars are consistent with the magnifications used and are clearly visible.

Scales showing 500 microns have been added to the newest version of Figure 2 and 3.

5. Present the data in Figures 4 and 8 using dN/dlogDp plots to normalize and log-scale particle size data. This will provide a clearer and more accurate representation of modal size ranges and size distributions.

Thank you for this suggestion. See below for the figures showing your recommendations. While logarithmic scaling is often useful for datasets spanning multiple orders of magnitude, we have chosen to maintain the linear scale presentation for several reasons:

1. Data Interpretability: Our particle size distributions primarily fall within a single order of magnitude (majority between 1-100 µm), making the linear scale more directly interpretable for readers. The logarithmic transformation (as shown in the alternative plots) compresses the distribution in a way that obscures important features of our data.

2. Physical Relevance: The linear scale better represents the actual physical differences in particle sizes that are relevant to hail formation processes. For example, in NG-1, the linear plot clearly shows the abundance of particles in the 0-50 µm range and the presence of larger particles extending to 250 µm, which is more difficult to discern in the logarithmic presentation.

3. Statistical Resolution: The linear scale better resolves the size distribution patterns within our most populated size ranges. For instance, in V-7, the linear histogram clearly shows the distribution of particles between 20-60 μm, which becomes compressed and less distinct in the logarithmic presentation.

The current linear presentation provides the most clear and physically meaningful representation of our particle size distributions, particularly for understanding the relationships between particle sizes in embryos versus outer layers.

[Figure]

V-7: Particle Size Distribution

[Figure]

NG-1: Particle Size Distribution

6. Increase the size of Figure 7a to make the axes and findings more legible. Additionally, enlarge the font size in Figure 7c to improve readability.

The figure has been updated with increased font sizes for better readability.

7. Include a brief discussion of the study's limitations, such as particle size detection thresholds and uncertainties in EDS composition analysis. Suggest directions for future research, such as using complementary techniques to confirm findings or expanding the geographical scope of the study.

Thank you for your insightful comment regarding the discussion of study limitations and future research directions. We have addressed these aspects in the manuscript as follows:
1. Study Limitations:
   - The manuscript discusses the particle size detection threshold, noting that our analysis is limited to particles approximately 1 μm and larger due to the resolution constraints of the Confocal Laser Scanning Microscope (CLSM) and Energy-Dispersive Spectroscopy (EDS).
   - We acknowledge uncertainties in EDS composition analysis, emphasizing that EDS is used primarily for element identification rather than quantification due to potential interferences and the complex morphology of particles.
2. Future Research Directions:
   - The manuscript suggests the integration of complementary techniques, such as Transmission Electron Microscopy (TEM) or Raman spectroscopy, to confirm and expand upon our findings.
   - We propose expanding the geographical scope of future studies to include hailstones from different regions and storm types, which could enhance our understanding of particle composition variability and sources.

These discussions are integrated into the "Discussion" and "Conclusion" sections, providing a overview of the study's scope and potential future research avenues.

---

## Author Response (AR2)

Line 40 and 499: In both instances work by Kiselev et al. (2017) is quoted for size-dependency of INPs and the role of an organic coating. Though the Kiselev et al. (2017) study is very nice, it does not focus on these points. Kiselev et al. (2017) examine ice nucleation using SEM. Their work may be cited better at another place in the document. Though I do not want to promote my own work here, we discuss size-dependency of INPs and organic matter as INP in our 2018 review (Knopf et al., ACS ESC, 2018). The AGU Monogragh review by Kanji et al. (2017) may cover these topics as well.

Thank you for your recommendation. The updates have been made on both lines 42 and 501. Additionally, the Kiselev et al. (2017) reference has been relocated to a more appropriate section at line 39.

Line 122: Missing citation?

The citation appears at the end of the paragraph because all the information presented here is drawn from that same reference.

Line 283 and following throughout the manuscript: In places there is a superfluous space after the bracket letter.

I intended to use the approximate symbol "~," but it was incorrectly coded in the libtext file. I have now corrected this and updated all occurrences throughout the manuscript.

Line 370: There is a question mark after "December".

I have updated the question mark to a period.

Line 451: There seems to be a citation issue: "(, USGS...)".

When I attempt to update this reference, it still appears unchanged. I will collaborate with the technical review team to ensure the citation is displayed correctly.

Line 531 and following: The headlines for the following four paragraphs are missing.

I have verified that the libtext file is coded correctly, but the titles still are not displayed in the PDF. I will collaborate closely with the technical review team to ensure the titles appear appropriately in each section.